# UTFC-DiffTracker: Short- and Long-Range Temporal Feature Consistency Diffusion for Underwater Object Tracking

## Abstract

Underwater object tracking (UOT) plays a significant role in marine animal protection, underwater search and rescue, and maritime security, yet faces distinctive challenges including color distortion, low visibility, similar distractors, and occlusion in complex environments. Existing approaches include frame-level trackers that employ enhancement-based or adaptation strategies, processing frames independently and leading to inconsistent feature styles and weakened temporal correlations. Video-level trackers leverage autoregressive mechanisms for temporal consistency but still struggle with persistent feature degradation and tracking drift in underwater environments. To overcome these limitations, this paper proposes UTFC-DiffTracker, the feature & video-level tracker that achieves spatiotemporal feature alignment. The framework integrates two core innovations: the Short-Range Temporal Feature Consistency integrates diffusion-based correction and dynamic style memory retention to resolve underwater feature degradation while maintaining temporal coherence; the Long-Range Temporal Feature Consistency enhances discrimination against distractors and occlusion through wavelet decomposition that separates historical tokens into stable structures and transient details. UTFC-DiffTracker achieves state-of-the-art performance on four UOT benchmarks while preserving semantic integrity and ensuring tracking reliability.

## 1 Introduction

Underwater object tracking (UOT) aims to uniquely identify and follow objects in video sequences across diverse underwater environments, given arbitrary target queries (Cai et al., 2023; González-Sabbagh & Robles-Kelly, 2023; Alawode et al., 2023). Unlike conventional open-air tracking, UOT faces unique challenges such as color distortion, low visibility, and the presence of visually similar distractors (Alawode et al., 2022; Zhang et al., 2024a). Early frame-level methods typically attempt to address these issues through enhancement-based preprocessing or domain adaptation (Luo et al., 2025; Zhang et al., 2025). However, processing each frame independently often yields inconsistent feature representations and weakens temporal coherence (Figure 1(a)). These approaches also struggle with persistent feature degradation caused by complex underwater conditions and limited annotated data, constraining the effectiveness of both enhancement and adaptation strategies.

To address the temporal consistency limitations of frame-level approaches, video-level trackers leverage autoregressive mechanisms and sequence-to-sequence learning to model dependencies across multiple frames, moving beyond traditional frame-pair matching (Wei et al., 2023; Zheng et al., 2024; Xie et al., 2024). Nevertheless, as shown in Figure 1(b), these methods still encounter challenges in underwater scenarios, including persistent feature degradation and tracking drift induced by color distortion, low visibility, and similar distractors. They often fail to effectively correct feature-level color shifts or maintain stable cross-frame coherence, ultimately limiting their robustness in complex underwater environments. These limitations highlight the need for more comprehensive solutions that jointly integrate feature-space correction with temporal alignment.

In this paper, we propose UTFC-DiffTracker, a feature- and video-level tracker that jointly performs feature-space correction while preserving temporal coherence (Figure 1(c)). This framework integrates two complementary modules: *Short-Range Temporal Feature Consistency*, which applies

Figure 1: Comparison of three tracking paradigms: (a) Frame-level trackers typically induce inconsistent feature styles and weak temporal correlations; (b) Video-level trackers still struggle with persistent feature degradation, which can lead to tracking drift; (c) Feature & Video-level tracker (Ours), aligns features across both space and time through a feature-aligned consistency framework, significantly improving tracking reliability in challenging underwater environments.

diffusion-based correction with dynamic style memory to mitigate feature degradation while maintaining frame-to-frame coherence; and *Long-Range Temporal Feature Consistency*, which leverages wavelet decomposition to disentangle stable structures from transient details, improving robustness to similar objects and occlusions. By combining these modules, UTFC-DiffTracker achieves spatiotemporal feature alignment that preserves semantic integrity and enhances tracking reliability in challenging underwater environments.

Our main contributions can be summarized as follows:

- We design the Short-Range Temporal Feature Consistency module that integrates diffusion-based correction with dynamic style memory retention to resolve underwater feature degradation while maintaining temporal coherence, preventing fragmentation from frame-independent processing.

- We develop the Long-Range Temporal Feature Consistency module that enhances discrimination against distractors and occlusion through wavelet decomposition, separating historical tokens into stable structures and transient details.

- We propose UTFC-DiffTracker, a feature-aligned consistency framework that achieves spatiotemporal feature alignment and establishes new state-of-the-art results on four UOT benchmarks, surpassing PUTracker by large margins (e.g., on UTB180, achieving 81.40 vs. 73.12 in AUC and 86.52 vs. 70.47 in Precision).

## 2 RELATED WORK

**Visual Object Tracking** Vision transformer-based trackers (Cui et al., 2022; Ye et al., 2022) unify feature extraction and interaction through one-stream encoders, while video-level approaches leverage autoregressive mechanisms for enhanced temporal modeling. (Xie et al., 2024) introduces learnable queries to capture appearance changes, (Wei et al., 2023) reframes tracking as coordinate sequence interpretation, and (Zheng et al., 2024) proposes token propagation for dense contextual association. These methods transition from frame-pair matching to sequence-to-sequence learning, improving temporal consistency through state propagation. However, in underwater scenarios with color distortion and similar distractors, they still struggle with feature-level color correction and cross-frame stability maintenance, limiting discrimination capability against persistent degradation.

**Underwater Object Tracking** Frame-level trackers address underwater degradation through enhancement-based methods like SiamFCA (Mei et al., 2024) with histogram equalization and ATCTrack (Luo et al., 2025) using SSIM thresholding, or adaptation-based approaches including OKTrack (Zhang et al., 2024a) employing knowledge distillation and PUTrack (Zhang et al., 2025) utilizing prompt-based fine-tuning. These methods process frames independently, causing inconsistent feature styles and weakened temporal correlations while failing to resolve persistent degradation due to complex underwater challenges and limited training data.

# 3 METHOD

## 3.1 OVERVIEW OF UTFC-DIFFTRACKER

Given a sequence of underwater visual clips $\mathcal{V} = \{(R_j, S_j)\}_{j=1}^L$, where $R_j, S_j \in \mathbb{R}^{3 \times H \times W}$ are the $j$-th reference and search frames, each pair $(R_j, S_j)$ is first partitioned into non-overlapping $P \times P$ patches and embedded into $D$-dimensional tokens via a patch embedding module $\mathcal{E}$, yielding $\dot{\mathcal{V}} = \mathcal{E}(\mathcal{V})$. The token sequence is then processed by $l$ Transformer layers to capture spatial and contextual relationships, producing refined features $\mathbf{X} = \mathcal{T}^l \circ \cdots \circ \mathcal{T}^1(\dot{\mathcal{V}}) \in \mathbb{R}^{N \times D}$, where $N$ is the total number of tokens across all frames. Target localization is performed by a lightweight dual-branch regression head $\mathcal{H}$, following ODTrack (Zheng et al., 2024), which predicts confidence scores through a classification branch and bounding box coordinates $\mathbf{B} = \mathcal{H}(\mathbf{X}) = [x_c, y_c, w, h]$ through a regression branch, where $(x_c, y_c)$ and $(w, h)$ denote the box center and dimensions. The model is trained with a composite loss $\mathcal{L} = \mathcal{L}_{cls} + \lambda_r \mathcal{L}_{reg} + \lambda_g \mathcal{L}_{GIoU}$, where $\lambda_r = 5$, $\lambda_g = 2$, $\mathcal{L}_{cls}$ employs focal loss (Lin et al., 2017) to mitigate class imbalance, and $\mathcal{L}_{reg}$ combines $L_1$ and GIoU losses (Rezatofighi et al., 2019) for precise target localization.

Tracking accuracy in underwater scenarios depends on the discriminative quality of the final search features $\mathbf{X}$ and the regression capability of $\mathcal{H}$. Since underwater degradation minimally affects $\mathbf{X}_s^l$ before it reaches $\mathcal{H}$, we focus on direct feature-space correction rather than pixel-level restoration. Feature-space correction is effective because $\mathbf{X}$ provides a compressed semantic encoding where degradation manifests as predictable distribution shifts, operating on condensed $N_s \times D$ token representations ($N_s \ll HW$). To improve temporal consistency, we propose UTCF-DiffTracker, a short- and long-range feature correction framework tailored for underwater scenarios. The short-range component performs diffusion-based feature correction to mitigate color distortion and low visibility while enforcing temporal coherence via dynamic style memory retention. The long-range component enhances discrimination against distractors and occlusions using Haar wavelet decomposition. This integrated approach, illustrated in Fig. 2(a), effectively addresses underwater feature degradation while maintaining temporal and semantic consistency.

## 3.2 DIFFUSION-BASED FEATURE CORRECTION

To implement short-range correction within our framework, we design a diffusion-based feature correction module. As illustrated in Figure 2(b), the process begins with an input sequence feature $\mathbf{X} \in \mathbb{R}^{B \times L \times C}$, where $B$ denotes the batch size, $L$ represents the sequence length, and $C$ is the feature dimension. This feature is first transformed through a multi-layer perceptron (MLP) to extract enriched representations, which involves a linear projection followed by a Gaussian error linear unit (GELU) activation function and layer normalization. The formula is given by: $\mathbf{H} = \text{LayerNorm}(\mathbf{W}_{p2} \cdot \text{GELU}(\mathbf{W}_{p1} \cdot \mathbf{X}))$ where $\mathbf{W}_{p1} \in \mathbb{R}^{C \times 2C}$ and $\mathbf{W}_{p2} \in \mathbb{R}^{2C \times C}$ are learnable weight matrices, and LayerNorm applies layer normalization to stabilize the features by normalizing across the feature dimension for each sample: $\text{LayerNorm}(\mathbf{x}) = \frac{\mathbf{x} - \mu}{\sigma} \odot \gamma + \beta$, with $\mu$ and $\sigma$ as the mean and standard deviation, and $\gamma$ and $\beta$ as learnable parameters. This step enhances feature expressiveness and stability for subsequent processing.

The sequence feature $\mathbf{H}$ is then used to generate a condition vector $\mathbf{c}$ that encapsulates global contextual information. This is achieved by first computing the mean of $\mathbf{H}$ along the sequence dimension to obtain a global representation, which is then passed through another MLP: $\mathbf{c} = \mathbf{W}_{c2} \cdot \text{GELU}\left(\mathbf{W}_{c1} \cdot \left(\frac{1}{L} \sum_{i=1}^L \mathbf{H}_i\right)\right)$ where $\mathbf{W}_{c1} \in \mathbb{R}^{C \times F}$ and $\mathbf{W}_{c2} \in \mathbb{R}^{F \times F}$ are learnable weights, and $F$ is the fusion dimension. The condition vector $\mathbf{c}$ serves to guide the diffusion process by providing a consistent reference of the sequence's overall style, which is crucial in object tracking to maintain temporal coherence against degradation like color distortion. The use of MLPs here introduces nonlinear transformations to capture complex patterns and interactions in the features.

For the diffusion process, a time step embedding $\mathbf{t}_{\text{emb}}$ is generated based on the diffusion step $t$ using sinusoidal encoding to represent temporal information in a continuous manner: $\mathbf{t}_{\text{emb}} = \text{LayerNorm}\left(\left[\sin\left(t \cdot e^{-k}\right), \cos\left(t \cdot e^{-k}\right)\right]_{k=0}^{D/2-1}\right)$ where $D$ is the embedding dimension, and LayerNorm ensures normalization. Simultaneously, a noise factor $\eta$ is computed using a cosine noise schedule to regulate the intensity of correction: $\eta = \cos\left(\frac{\pi}{2} \cdot \frac{t+s}{T+s}\right)$ with $T$ as the maximum

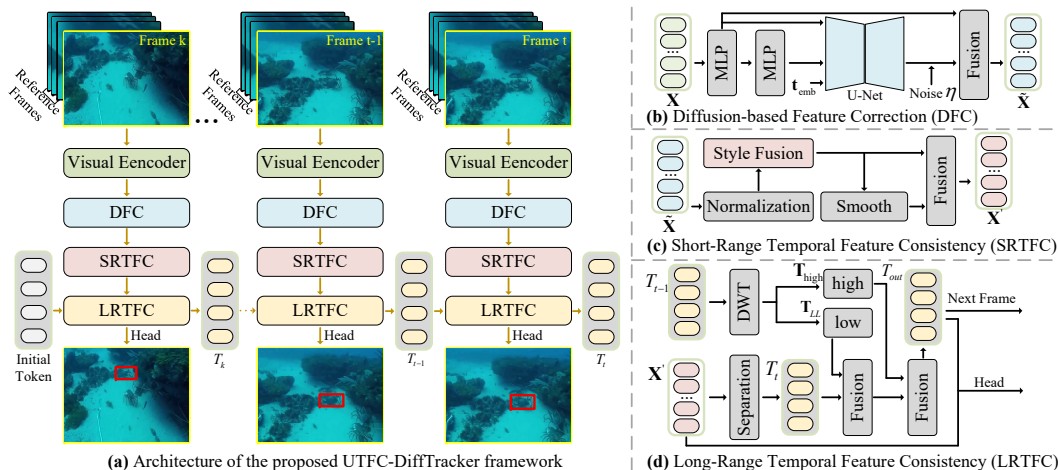

Figure 2: Architecture of the proposed UTFC-DiffTracker framework integrating complementary temporal consistency mechanisms. **(a)** illustrates the overall framework with three core components. **(b)** Diffusion-based Feature Correction (DFC) adaptively transforms features to mitigate color distortion and low visibility. **(c)** Short-Range Temporal Feature Consistency (SRTFC) maintains feature coherence across consecutive frames via dynamic style memory.**(d)** Long-Range Temporal Feature Consistency (LRTFC) enhances discrimination against distractors and occlusions using wavelet decomposition, capturing both structural persistence and transient details.

diffusion steps (e.g., $T = 700$), and $s$ as a small constant for numerical stability. The noise factor $\eta$ controls the amount of noise attenuation, where higher $t$ values lead to finer correction, while the time step embedding $\mathbf{t}_{\text{emb}}$ provides the diffusion time context to the network; both are derived from $t$ but serve distinct roles.

The core correction is performed by a U-Net architecture, which takes $\mathbf{H}$, $\mathbf{t}_{\text{emb}}$, and $\mathbf{c}$ as inputs to produce a corrected feature $\mathbf{H}_{\text{corr}}$. This U-Net employs normalization to inject condition information and handles feature refinement through down-sampling and up-sampling paths. The process approximates the reverse diffusion step, and theoretically, it can be described by the formula:

$$\mathbf{H}_{t-1} = \frac{1}{\sqrt{\alpha_t}} \left( \mathbf{H}_t - \frac{1 - \alpha_t}{\sqrt{1 - \gamma_t}} f_\theta(\mathbf{c}, \mathbf{H}_t, t) \right) + \lambda_t \beta_t \epsilon_t \tag{1}$$

where $\alpha_t$, $\gamma_t$, and $\beta_t$ are parameters of the diffusion schedule, $\epsilon_t \sim \mathcal{N}(0, \mathbf{I})$ is Gaussian noise with $\mathbf{I}$ denoting the identity matrix representing the covariance structure of isotropic noise, and $f_\theta$ represents the U-Net function that predicts noise for denoising. In practice, the U-Net output is directly used as $\mathbf{H}_{\text{corr}} = f_\theta(\mathbf{H}, \mathbf{t}_{\text{emb}}, \mathbf{c})$.

To enhance robustness, noise factor $\eta$ is added to the corrected feature, simulating the stochasticity of reverse diffusion step:

$$\mathbf{H}'_{\text{corr}} = \mathbf{H}_{\text{corr}} \cdot (1 + \eta) + \lambda \cdot \epsilon \tag{2}$$

where $\epsilon$ and $\lambda$ are learnable residual weights. Noise addition introduces variability to handle uncertainties in underwater environments, such as low visibility or distractors.

Finally, the noise-added corrected feature is fused with the original sequence feature $\mathbf{H}$ to produce the output. A fusion weight is generated by an MLP with sigmoid activation: $\mathbf{g} = \sigma \left( \mathbf{W}_g \cdot [\mathbf{H} \oplus \mathbf{H}'_{\text{corr}}] \right)$, where $\mathbf{W}_g \in \mathbb{R}^{2C \times C}$ is a weight matrix, $\oplus$ denotes concatenation along the feature dimension, and $\sigma$ is the sigmoid function. The fused output is:

$$\widetilde{\mathbf{X}} = \mathbf{g} \odot \mathbf{H}'_{\text{corr}} + (1 - \mathbf{g}) \odot \mathbf{H} \tag{3}$$

The fusion process combines original and corrected features through a learnable weighting scheme, retaining beneficial aspects of both to balance consistency and enhancement for reliable tracking. The diffusion step t controls correction granularity via noise scheduling and time embeddings, with higher steps enabling refined feature reconstruction against underwater degradation. For more detailed implementation specifics, refer to the pseudocode in Algorithm 1.

### 3.3 SHORT-RANGE TEMPORAL FEATURE CONSISTENCY

The Short-Range Temporal Feature Consistency (SRTFC) module addresses temporal style inconsistencies in underwater object tracking by leveraging historical style information to enhance feature coherence, as illustrated in the framework overview in Figure 2(c). The process begins with the input of original features $\mathbf{X}$ and diffusion-corrected features $\widetilde{\mathbf{X}}$, producing style-aligned features $\mathbf{X}'$.

First, the current style vector $\mathbf{s}_{\text{curr}} \in \mathbb{R}^{B \times C}$ is extracted from $\mathbf{X}$ by computing the mean over the sequence dimension: $\boldsymbol{\mu} = \frac{1}{L} \sum_{l=1}^{L} \mathbf{X}_{:,l,:}$, followed by a linear transformation with weight matrix $\mathbf{W}_1 \in \mathbb{R}^{C \times C}$ and bias $\mathbf{b}_1 \in \mathbb{R}^C$, and a ReLU activation, resulting in $\mathbf{s}_{\text{curr}} = \text{ReLU}(\boldsymbol{\mu}\mathbf{W}_1 + \mathbf{b}_1)$. This vector captures the dominant stylistic characteristics of the current input.

A dynamic memory matrix $\mathbf{M} \in \mathbb{R}^{M \times C}$, where $M$ is the memory size, stores historical style vectors to maintain temporal consistency by retaining recent style information; it is updated by replacing the oldest entry with $\mathbf{s}_{\text{curr}}$, ensuring the memory contains the most relevant styles for fusion. To fuse historical styles, the cosine similarity between $\mathbf{s}_{\text{curr}}$ and each historical style $\mathbf{s}^{(k)}$ in $\mathbf{M}$ is computed as $\cos(\mathbf{s}_{\text{curr}}, \mathbf{s}^{(k)}) = \frac{\mathbf{s}_{\text{curr}} \cdot \mathbf{s}^{(k)}}{\|\mathbf{s}_{\text{curr}}\| \|\mathbf{s}^{(k)}\|}$, and the top-$K$ most similar styles are selected. Their weights are obtained using a softmax function:

$$\omega_k = \frac{\exp(\cos(\mathbf{s}_{\text{curr}}, \mathbf{s}^{(k)}))}{\sum_{j=1}^{K} \exp(\cos(\mathbf{s}_{\text{curr}}, \mathbf{s}^{(j)}))}, \tag{4}$$

and the fused style vector is computed as $\mathbf{s}_{\text{fused}} = \sum_{k=1}^{K} \omega_k \mathbf{s}^{(k)}$.

Next, style adaptive normalization is applied to $\widetilde{\mathbf{X}}$ to align it with the fused style. Instance normalization is performed per channel: for each batch $b$ and channel $c$, the mean $\mu_{b,c} = \frac{1}{L} \sum_{l=1}^{L} \widetilde{X}_{b,l,c}$ and variance $\sigma_{b,c}^2 = \frac{1}{L} \sum_{l=1}^{L} (\widetilde{X}_{b,l,c} - \mu_{b,c})^2$ are computed, and the normalized features are given by: $\hat{X}_{\text{norm},b,l,c} = \frac{\widetilde{X}_{b,l,c} - \mu_{b,c}}{\sqrt{\sigma_{b,c}^2 + \epsilon}} \cdot \gamma_c + \beta_c$, where $\gamma_c$ and $\beta_c$ are learnable parameters. These normalized features are then combined with $\mathbf{s}_{\text{fused}}$ through linear interpolation to produce style-transferred features $\mathbf{X}_{\text{transferred}}$, though the exact interpolation mechanism is omitted here to avoid gate terminology.

Temporal smoothing is applied to enhance coherence across sequences using a 1D convolution with a kernel size of 3 and padding of 1, implemented with a learnable kernel $\mathbf{K} \in \mathbb{R}^3$. The smoothed features are computed as:

$$X_{\text{smooth},b,l,c} = \sum_{d=-1}^{1} K_{d+1} \cdot X_{\text{transferred},b,l+d,c}, \tag{5}$$

with appropriate padding to maintain sequence length. Finally, the output features are formed by a weighted sum:

$$\mathbf{X}' = \alpha \mathbf{X}_{\text{transferred}} + (1 - \alpha) \mathbf{X}_{\text{smooth}}, \tag{6}$$

where $\alpha = 0.7$ balances feature integrity and temporal smoothness. This results in features that are both style-consistent and coherent across frames. For more detailed implementation specifics, refer to the pseudocode in Algorithm 2.

### 3.4 LONG-RANGE TEMPORAL FEATURE CONSISTENCY

This work is based on ODTrack (Zheng et al., 2024), which proposes an online dense temporal token learning framework for visual tracking. The LTRFC enhances temporal consistency by propagating token sequences across frames in an auto-regressive manner, as illustrated in the framework overview in Figure 2(d). The input to the module consists of the previous token sequence $\mathbf{T}_{t-1} \in \mathbb{R}^{B \times N \times C}$ and the current token sequence $\mathbf{T}_t \in \mathbb{R}^{B \times N \times C}$, where $B$ is the batch size, $N$ is the number of tokens, and $C$ is the feature dimension. The current token $\mathbf{T}_t$ is derived directly from the transformer encoder output of the current frame, specifically by selecting the first $L$ tokens along the sequence dimension, representing the target instance's features.

To ensure compatibility with the wavelet transformation, both token sequences are padded to a minimum length of $N_{\min} = 4$ tokens if $N < 4$, using zero-padding along the token dimension.

The previous token $\mathbf{T}_{t-1}$ is then reshaped into a 2D format $\mathbf{T}_{t-1}^{\text{2D}} \in \mathbb{R}^{B \times C \times H \times W}$, where $H \times W = N$, to facilitate spatial processing. This reshaped tensor undergoes a 2D discrete wavelet transform (DWT) using the Haar wavelet, which decomposes it into a low-frequency component $\mathbf{T}_{LL} \in \mathbb{R}^{B \times C \times H/2 \times W/2}$ and high-frequency components $\mathbf{T}_{LH}, \mathbf{T}_{HL}, \mathbf{T}_{HH} \in \mathbb{R}^{B \times C \times H/2 \times W/2}$. These components are then reshaped back to the token format $\mathbb{R}^{B \times N' \times C}$, where $N' = (H/2) \times (W/2)$, and truncated to the original token length $N$ by selecting the first $N$ tokens.

The fusion process begins by concatenating the low-frequency component $\mathbf{T}_{LL}$, the high-frequency components $\mathbf{T}_{LH}, \mathbf{T}_{HL}, \mathbf{T}_{HH}$, and the current token $\mathbf{T}_t$ along the feature dimension, forming a combined tensor $\mathbf{X} = [\mathbf{T}_{LL}, \mathbf{T}_{LH}, \mathbf{T}_{HL}, \mathbf{T}_{HH}, \mathbf{T}_t] \in \mathbb{R}^{B \times N \times 5C}$. This tensor is passed through a linear transformation followed by a sigmoid activation function to generate fusion weights. Specifically, $\mathbf{G} = \sigma\left(\mathbf{W}_g \mathbf{X} + \mathbf{b}_g\right)$, where $\mathbf{W}_g \in \mathbb{R}^{5C \times 2C}$ is a learnable weight matrix, $\mathbf{b}_g \in \mathbb{R}^{2C}$ is a bias term, and $\sigma$ denotes the sigmoid function. The output $\mathbf{G} \in \mathbb{R}^{B \times N \times 2C}$ is split into two tensors along the feature dimension: $\mathbf{G}_1 \in \mathbb{R}^{B \times N \times C}$ and $\mathbf{G}_2 \in \mathbb{R}^{B \times N \times C}$.

The high-frequency components are summed to form a consolidated high-frequency representation $\mathbf{T}_{\text{HF}} = \mathbf{T}_{LH} + \mathbf{T}_{HL} + \mathbf{T}_{HH}$. This is then modulated by the first weight tensor $\mathbf{G}_1$ through element-wise multiplication: $\mathbf{T}_{\text{high}} = \mathbf{T}_{\text{HF}} \odot \mathbf{G}_1$. Similarly, the low-frequency component $\mathbf{T}_{LL}$ and the current token $\mathbf{T}_t$ are combined using the second weight tensor $\mathbf{G}_2$:

$$\mathbf{T}_{\text{new\_ll}} = \mathbf{T}_{LL} \odot \mathbf{G}_2 + \mathbf{T}_t \odot (\mathbf{1} - \mathbf{G}_2), \tag{7}$$

where $\mathbf{1}$ denotes a tensor of ones with the same dimensions as $\mathbf{G}_2$. The results are concatenated along the feature dimension to form $\mathbf{T}_{\text{fused}} = [\mathbf{T}_{\text{new\_ll}}, \mathbf{T}_{\text{high}}] \in \mathbb{R}^{B \times N \times 2C}$. Finally, this fused tensor is projected back to the original feature dimension $C$ through a linear layer: $\mathbf{T}_{\text{out}} = \mathbf{W}_p \mathbf{T}_{\text{fused}} + \mathbf{b}_p$, where $\mathbf{W}_p \in \mathbb{R}^{2C \times C}$ is a learnable weight matrix and $\mathbf{b}_p \in \mathbb{R}^C$ is a bias term. The output $\mathbf{T}_{\text{out}} \in \mathbb{R}^{B \times N \times C}$ serves as the propagated token sequence for the next frame, ensuring temporal coherence. For more detailed implementation specifics, refer to the pseudocode in Algorithm 3.

# 4 EXPERIMENTS

## 4.1 IMPLEMENTATION DETAILS

**Training.** The model training employs the ViT-Base (Dosovitskiy et al., 2021) architecture initialized with MAE (He et al., 2022) pre-training parameters. To address the scarcity of underwater training data, we reconfigure datasets to LaSOT (Fan et al., 2019), WebUOT-1M (Zhang et al., 2024a), HUT290 (Zhang et al., 2025), and UVOT400 (Alawode et al., 2023), implementing UOSTrack's (Li et al., 2023) hybrid training strategy. The input with template frames resized to 128×128 pixels and search regions to 256×256 pixels. Optimization is performed using the AdamW optimizer with learning rates set to $1 \times 10^{-5}$ for the backbone and $1 \times 10^{-4}$ for other components, accompanied by $1 \times 10^{-4}$ weight decay regularization. The learning rate drops by a factor of 10 after 240 epochs. Training spans 300 epochs with random batch sampling, executed on a server equipped with three 24 GB NVIDIA GeForce 4090 GPUs using a batch size of 5.

**Inference.** During inference, UTFC-DiffTracker processes video sequences through its feature correction pipeline, maintaining temporal coherence in dynamic underwater environments. All inference runs on a single NVIDIA GeForce 4090 GPU.

## 4.2 COMPARISON WITH THE STATE-OF-THE-ARTS

We compare UTFC-DiffTracker against state-of-the-art underwater trackers, including PUTrack (Zhang et al., 2025), OKTrack (Zhang et al., 2024a), and UOSTrack (Li et al., 2023), and representative VOT trackers such as OSTrack (Ye et al., 2022), ODTrack (Zheng et al., 2024), and MCITrack (Kang et al., 2025). This comparison evaluates performance across diverse challenges.

**Quantitative comparison.** UTFC-DiffTracker achieves state-of-the-art performance across four UOT benchmarks in Table 1, showing major gains over existing methods. On WebUOT-1M (Zhang et al., 2024a), it scores 67.62% AUC, beating ODTrack by 3.80%. Similarly, it reaches 61.5% AUC on UW-COT220 (Zhang et al., 2024b), exceeding ODTrack by 1.12 points. It keeps this lead on VMAT (Cai et al., 2023) with 61.25% AUC and UTB180 (Alawode et al., 2022) with 81.40% AUC,

Table 1: Comparison of the area under the curve (AUC), precision (P) metrics, and average FPS with state-of-the-art methods on four UOT benchmarks: WebUOT-1M, UW-COT220, VMAT, and UTB180. Asterisk (*) denotes trackers specifically designed for UOT. The top two performers in each metric category are highlighted in red and blue respectively.

| Method | WebUOT-1M | | UW-COT220 | | VMAT | | UTB180 | | FPS |
|---|---|---|---|---|---|---|---|---|---|
| | AUC | P | AUC | P | AUC | P | AUC | P | |
| **UTFC-DiffTracker*** | 67.62 | 68.84 | 61.50 | 58.35 | 61.25 | 78.07 | 81.40 | 86.52 | 32.35 |
| MCITrack (Kang et al., 2025) | 62.02 | 60.44 | 60.10 | 57.24 | 60.89 | 74.92 | 65.37 | 63.68 | 24.28 |
| SGLATrack (Xue et al., 2025) | 53.73 | 46.70 | 46.77 | 37.00 | 45.75 | 55.74 | 59.41 | 49.58 | 151.4 |
| ORTrack (Wu et al., 2025) | 52.17 | 45.12 | 45.16 | 35.17 | 44.49 | 55.21 | 56.66 | 47.20 | 109.7 |
| PUTrack* (Zhang et al., 2025) | 54.06 | 49.56 | 47.05 | 38.54 | 43.14 | 54.12 | 73.12 | 70.47 | 28.17 |
| ODTrack (Zheng et al., 2024) | 63.82 | 60.88 | 60.38 | 56.38 | 54.66 | 67.73 | 69.96 | 64.48 | 34.42 |
| OKTrack* (Zhang et al., 2024a) | 62.05 | 59.13 | 59.17 | 53.82 | 53.82 | 66.83 | 69.24 | 66.46 | 69.78 |
| HIPTrack (Cai et al., 2024) | 61.58 | 58.01 | 57.90 | 53.63 | 55.83 | 68.96 | 66.14 | 60.65 | 92.14 |
| ARTrack(Wei et al., 2023) | 62.44 | 59.69 | 60.02 | 55.89 | 58.67 | 74.17 | 68.80 | 65.02 | 37.31 |
| SeqTrack (Chen et al., 2023) | 57.63 | 54.56 | 55.95 | 50.64 | 55.67 | 69.97 | 60.73 | 55.25 | 28.09 |
| GRM (Gao et al., 2023) | 58.86 | 55.35 | 56.14 | 50.12 | 57.63 | 71.68 | 63.99 | 58.62 | 35.16 |
| UOSTrack* (Li et al., 2023) | 60.57 | 55.96 | 56.87 | 50.94 | 55.84 | 67.90 | 66.65 | 60.05 | 70.08 |
| DropTrack (Wu et al., 2023) | 59.94 | 56.31 | 57.03 | 52.36 | 55.82 | 68.90 | 63.47 | 58.32 | 27.25 |
| OSTRack (Ye et al., 2022) | 58.50 | 54.35 | 53.85 | 47.26 | 52.78 | 65.92 | 62.51 | 55.90 | 73.91 |
| TransT (Chen et al., 2021) | 54.60 | 49.84 | 49.34 | 41.62 | 43.52 | 52.94 | 57.61 | 50.79 | 44.46 |

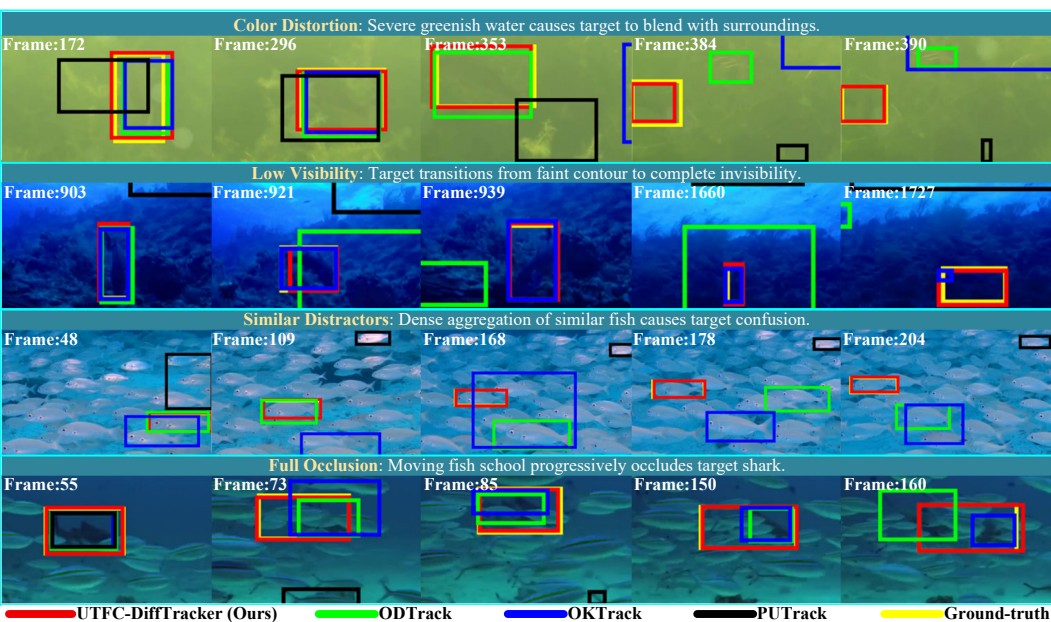

Figure 3: Qualitative results on WebUOT-1M demonstrate robustness against key underwater challenges: color distortion, low visibility, similar distractors, and full occlusion.

topping rivals by 0.36 and 8.28 points. The framework demonstrates robust performance across diverse underwater conditions; although marginally slower, it maintains real-time operation.

**Attributes comparison.** Figure 4 validates UTFC-DiffTracker's exceptional capability in key underwater challenges on WebUOT-1M. For color distortion, it achieves 78.7% AUC in Light Brown water color variation and 72.2% in Light Yellow. In low visibility conditions, it attains 62.9% AUC. Against similar distractors, it reaches 60.9% AUC. For occlusion challenges, it achieves 47.7% AUC in full occlusion. Additional strengths include 63.7% AUC against camouflage and 62.2% AUC in fast motion, confirming comprehensive handling of underwater complexities.

Table 2: We conduct a module ablation study on the VMAT benchmark. Top-2 performers in AUC, P, Params, and FPS metrics are highlighted in red and blue. Hybrid Training Strategy (HTS) refers to training dataset replacement. The Diffusion-based Feature Correction module (DFC), Short-Range Temporal Feature Consistency (SRTFC), and Long-Range Temporal Feature Consistency (LRTFC) constitute proposed modules with SRTFC dependent on DFC.

|  | HTS | DFC | SRTFC | LRTFC | AUC | P | Params | FPS |
|---|---|---|---|---|---|---|---|---|
| ① | ✗ | ✗ | ✗ | ✗ | 54.66 | 67.73 | 92.12M | 34.42 |
| ② | ✓ | ✗ | ✗ | ✗ | 55.54 | 68.86 | 92.12M | 34.42 |
| ③ | ✓ | ✓ | ✗ | ✗ | 57.12 | 71.65 | 96.92M | 33.12 |
| ④ | ✓ | ✗ | ✗ | ✓ | 56.62 | 70.17 | 92.12M | 34.42 |
| ⑤ | ✓ | ✓ | ✓ | ✗ | 58.73 | 73.70 | 98.70M | 32.35 |
| ⑥ | ✓ | ✓ | ✗ | ✓ | 57.57 | 71.80 | 96.92M | 33.12 |
| ⑦ | ✓ | ✓ | ✓ | ✓ | 61.25 | 78.07 | 98.70M | 32.35 |

**Qualitative comparison.** Visual comparison with ODTrack, PUTrack, and OKTrack on WebUOT-1M shows our robustness. In severe greenish water distortion, UTFC-DiffTracker keeps precise tracking while others lose targets at frame 384. Under low visibility with invisible targets, UTFC-DiffTracker tracks until frame 1727 while others fail before. Against dense, similar distractors, UTFC-DiffTracker accurately distinguishes targets to frame 204 while others confuse targets. During full fish occlusion, UTFC-DiffTracker recovers fast post-occlusion while others show tracking drift and scale errors. These results prove superior adaptability in key underwater challenges.

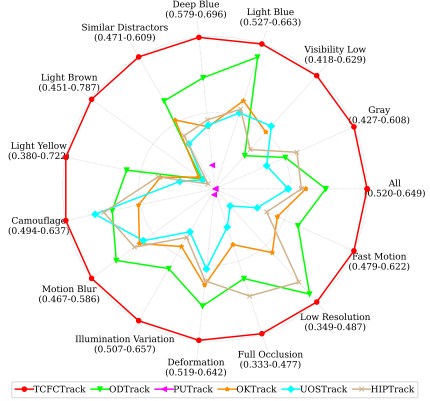

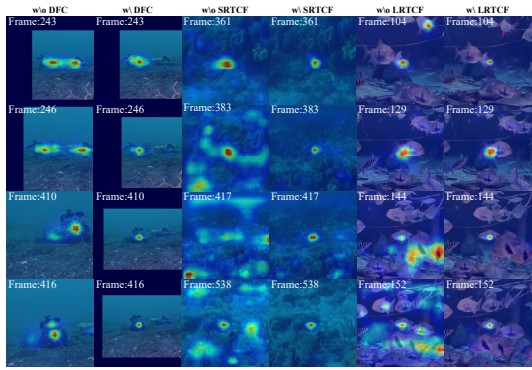

Figure 4: AUC of different attributes on WebUOT-1M.

Figure 5: The attention map of DFC, SRTFC, and LRTFC. Comparison of attention maps. Note that w/o or w/ denote the tracker without and with DFC, SRTFC, or LRTFC.

## 4.3 IMPROVEMENT OF BASELINE

Module ablation experiments on the VMAT benchmark using ODTrack (Zheng et al., 2024) as baseline (①) reveal progressive performance gains. The ② incorporating a Hybrid Training Strategy, achieves 55.54% AUC, demonstrating 0.88% improvement through domain adaptation. The ③ adding a Diffusion-based Feature Correction, gains 1.58% AUC to 57.12%, resolving color distortion and low visibility. The ④ integrating Long-Range Temporal Feature Consistency shows 56.62% AUC, providing 0.96% improvement against distractors. The ⑤ combining diffusion correction with Short-Range Temporal Feature Consistency, achieves 58.73% AUC, 1.61% gain through temporal coherence maintenance. The ⑥ merging diffusion correction with long-range consistency reaches 57.57% AUC. The full UTFC-DiffTracker (⑦) achieves 61.25% AUC, +6.59 point gain, with 32.35 FPS. Although the diffusion steps increase computation, it still maintains real-time tracking.

**Visualization of attention.** Figure 5 demonstrates attention refinement across the three proposed modules: DFC counteracts color distortion by reducing non-target dispersion and sharpening bound-

ary focus; SRTFC maintains temporal coherence by stabilizing attention weights across consecutive frames; LRTFC suppresses background distractors through wavelet-based structural preservation. Collectively, these mechanisms enhance spatiotemporal attention alignment, significantly improving tracking robustness in challenging underwater environments, as further evidenced in Figure 8.

## 4.4 ABLATION STUDIES

Table 3: Comparison of DFC's diffusion steps.

| Steps | AUC | P |
|---|---|---|
| 600 | 60.01 | 75.92 |
| 700 | 61.25 | 78.07 |
| 800 | 60.11 | 76.07 |
| 900 | 59.60 | 75.55 |

Table 4: Comparison of SRTFC's update strategy.

| Strategy | AUC | P |
|---|---|---|
| *FIFO* | 61.25 | 78.07 |
| *Ema* | 60.33 | 76.61 |
| *Mean* | 59.24 | 75.2 |
| *Random* | 60.68 | 76.91 |

Table 5: Comparison of LRTFC's separation.

| Method | AUC | P |
|---|---|---|
| *Haar* | 61.25 | 78.07 |
| *Fourier* | 60.71 | 77.15 |
| *Attention* | 60.76 | 77.11 |
| *Inception* | 60.31 | 76.74 |

**Diffusion steps study.** This investigation examines the impact of diffusion step selection on tracking performance, emphasizing the critical role of step calibration in balancing feature refinement and computational efficiency. As demonstrated in Table 3, optimal results are achieved at 700 steps with an AUC of 61.25%, where the diffusion process corrects underwater degradation through sufficient noise attenuation and feature reconstruction. Performance declines to 60.01% AUC at 600 steps due to inadequate refinement, as the shorter schedule fails to address color distortion and low visibility. Increasing steps to 800 and 900 further reduces AUC to 60.11% and 59.60%, respectively, resulting from over-smoothing that erodes discriminative details and increases computational burden.

**SRTFC's update strategy.** This analysis examines historical style update strategies for the SRTFC module's dynamic memory matrix, demonstrating that FIFO's rotating buffer replacement achieves optimal temporal coherence with 61.25% AUC by preserving recent degradation patterns while discarding obsolete data. This approach outperforms exponential moving averaging (60.33% AUC), which oversmooths temporal variations, global mean aggregation (59.24% AUC) that dilutes distinctive style characteristics, and random replacement (60.68% AUC), which introduces representation inconsistencies. The rotating buffer mechanism effectively maintains style persistence across consecutive frames, providing robust feature alignment against underwater challenges, including color distortion and illumination variations, through selective retention of relevant stylistic information.

**LRTFC's separation method.** This study tests four feature separation methods in LRTFC on VMAT. *Haar* uses wavelet decomposition to split historical tokens into stable structures and transient details, clearly keeping complementary data. *Fourier* employs spectral masking to isolate frequency parts. *Attention* applies spatial attention for component separation. *Inception* uses fixed convolutional kernels to get multi-scale features. As shown in Table 5, *Haar* scores best with 61.25% AUC, topping *Fourier* by 0.54%, *Attention* by 0.49%, and *Inception* by 0.94%. These findings prove that clear separation of persistent structures and transient details via wavelet analysis best boosts discrimination against distractors and occlusion, solving issues in standard temporal methods.

## 5 CONCLUSION

This paper presents UTFC-DiffTracker, a feature-aligned consistency framework designed for UOT that achieves spatiotemporal feature alignment. The framework integrates two core components: the Short-Range Temporal Feature Consistency module, which employs diffusion-based correction with dynamic style memory retention to resolve feature degradation while maintaining temporal coherence, and the Long-Range Temporal Feature Consistency module, which utilizes wavelet decomposition to enhance discrimination against distractors and occlusion by separating historical tokens into stable structures and transient details. This approach addresses feature degradation while preserving spatiotemporal consistency, demonstrating competitive performance on four UOT benchmarks.

## ETHICS STATEMENT

This work adheres to the ICLR Code of Ethics and was conducted in accordance with its principles of responsible stewardship, scientific excellence, and social benefit. Our research on underwater object tracking demonstrates positive potential applications in marine animal protection, underwater search and rescue, and maritime security—areas that align with the Code's principle of contributing to society and human well-being.

We have carefully considered the potential ethical implications of our work, particularly regarding environmental impact and data usage. The research utilizes existing benchmark datasets, including WebUOT-1M, UW-COT220, VMAT, and UTB180, which incorporate video data from online platforms and marine observatories. All data were used in accordance with their original licenses and intended research purposes. No new data collection involving human subjects or animal experimentation was conducted specifically for this study.

Our methodology emphasizes transparency and reproducibility, with detailed descriptions of architectures, training procedures, and evaluation metrics provided to facilitate verification and build upon our work. We have implemented measures to minimize potential negative consequences, including rigorous validation against similar distractors and occlusion scenarios to reduce false tracking that could impact marine conservation efforts.

The research was conducted with consideration for potential dual-use implications. While developed for beneficial applications, tracking technologies could potentially be misused. We have deliberately focused on open research and transparency as a mitigation strategy, believing that responsible disclosure and community engagement provide the best safeguard against misuse.

We acknowledge the support from our institutional resources and confirm that no conflicts of interest influenced this work. All researchers involved in this project have completed ethics training relevant to their respective institutions, and the research was conducted in compliance with applicable institutional and national guidelines.

## REPRODUCIBILITY STATEMENT

We have made extensive efforts to ensure the reproducibility of our work. The core methodology, including the diffusion-based feature correction and temporal consistency modules, is detailed in Section 3 of the main text. Full algorithm descriptions and pseudocode for all novel components are provided in Algorithms 1, 2, and 3 in the Appendix C. Comprehensive implementation details, such as computing environments, training parameters, and inference settings, are outlined in Appendix B. The benchmark datasets used for evaluation are thoroughly documented in Appendix B.2, including data sources and processing steps. Additional experimental results, ablation studies, and visualizations are included in Appendices E and related sections.

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

## A   LARGE LANGUAGE MODELS USAGE

In accordance with the ICLR 2026 policy on Large Language Models (LLMs), we provide this statement regarding the use of LLMs in the preparation of this work. LLMs were employed solely as a general-purpose assist tool during the research and writing process, primarily in two specific aspects:

- **Writing Assistance and Polishing:** LLMs were used to assist with language polishing and refinement of certain sections of the manuscript. This included improving sentence structure, enhancing clarity of expression, and ensuring consistent academic tone throughout the paper. All conceptual contributions, technical explanations, and scientific content remain the original work of the authors.

- **Code Debugging Assistance:** During the implementation of our proposed UTFC-DiffTracker framework, LLMs were occasionally consulted to help identify and resolve routine programming issues and syntax errors.

The LLMs were not involved in the formulation of research ideas, the design of methodologies, or the theoretical analysis. The core scientific contributions, including the short-range and long-range temporal feature consistency mechanisms, the diffusion-based correction approach, and all experimental designs, are entirely the product of the authors' intellectual work.

We affirm that all authors take full responsibility for the entire content of this paper, including any portions that may have been refined with LLM assistance. We have carefully reviewed and verified all content to ensure its accuracy and originality, and to prevent any potential issues of plagiarism or scientific misconduct.

## B   DETAILED IMPLEMENTATION

### B.1   SETTINGS

**Computing Environments.**   All experiments run on an Ubuntu 24.04.2 LTS server equipped with dual Intel Xeon Gold 6330N CPUs and three NVIDIA GeForce RTX 4090 GPUs, each providing 24GB memory, supported by 1024GB RAM. The software environment utilizes Python 3.8.20, PyTorch 1.10.0, and CUDA 11.3 for computational tasks.

**Training.**   The model training employs the ViT-Base Dosovitskiy et al. (2021) architecture initialized with MAE He et al. (2022) pre-training parameters. To address the scarcity of underwater training data, we reconfigure datasets to LaSOT Fan et al. (2019), WebUOT-1M Zhang et al. (2024a), HUT290 Zhang et al. (2025), and UVOT400 Alawode et al. (2023), implementing UOSTrack's Li et al. (2023) hybrid training strategy. The input configuration uses 3 template frames resized to 128×128 pixels and 2 search frames resized to 256×256 pixels per sample. We randomly sample 60,000 image pairs each epoch. Optimization employs the AdamW optimizer with learning rates set to $1 \times 10^{-5}$ for the backbone and $1 \times 10^{-4}$ for other components, accompanied by $1 \times 10^{-4}$ weight decay regularization. The learning rate drops by a factor of 10 after 240 epochs. Training spans 300 epochs, executed on a server equipped with three 24 GB NVIDIA GeForce 4090 GPUs using a batch size of 5.

**Inference**   During inference operations, UTFC-DiffTracker processes sequential video frames through its integrated feature correction pipeline, maintaining temporal coherence while adapting to dynamic underwater conditions. All inference procedures are executed on a single NVIDIA GeForce 4090 GPU. This streamlined implementation achieves robust target localization through multi-stage feature refinement while minimizing computational overhead.

### B.2   BENCHMARK DATASETS

Four specialized benchmarks advance underwater object tracking (UOT) research. WebUOT-1M provides large-scale multi-modal tracking data. UW-COT220 focuses on camouflaged objects with segmentation annotations. VMAT enables semi-supervised marine animal tracking. UTB180 introduces underwater-specific evaluation attributes. These datasets collectively address critical challenges in UOT.

**WebUOT-1M**   WebUOT-1M Zhang et al. (2024a) is the largest underwater tracking benchmark. It contains 1.1 million frames across 1,500 video clips. These clips come from online platforms like YouTube and BiliBili. The dataset covers 408 marine categories grouped into 12 superclasses. Professional annotation provides high-quality bounding boxes. Language prompts enable multi-modal tracking research. This benchmark exceeds previous datasets in scale and diversity.

**UW-COT220**   UW-COT220 Zhang et al. (2024b) is the first multi-modal benchmark for underwater camouflaged object tracking. It includes 220 video sequences with 159,000 frames. These sequences cover 96 marine species. Professional annotation provides bounding boxes and segmentation masks. Each frame has absence labels for occlusion cases. Videos come from online platforms and existing datasets. Quality control ensures accurate annotations. This benchmark advances underwater vision-language research.

**VMAT**   The VMAT dataset Cai et al. (2023) is the first semi-supervised marine animal tracking benchmark. It contains 33 video sequences with 74,000 frames. Data collection used divers and autonomous vehicles. Sequences cover 17 marine species across diverse habitats. Professional annotation provides dense bounding boxes. The dataset features unique underwater challenges.

Marine-specific attributes complement standard tracking attributes. This benchmark enables authentic algorithm evaluation.

**UTB180** The UTB180 benchmark Alawode et al. (2022) provides comprehensive underwater tracking data. It contains 180 sequences with 58,000 annotated frames. These sequences capture diverse marine species. Data sources include online platforms and marine observatories. Frame resolutions range from 1520×2704 to 959×1277 pixels. Rigorous five-stage annotation ensures quality. The benchmark introduces ten underwater-specific attributes. These attributes evaluate tracking robustness in challenging conditions.

# C    ALGORITHMS

The UTFC-DiffTracker framework integrates three core algorithmic components to address underwater tracking challenges. Algorithm 1 performs Diffusion-based Feature Correction to resolve color distortion and low visibility. Algorithm 2 ensures Short-Range Temporal Feature Consistency through dynamic style retention. Algorithm 3 achieves Long-Range Temporal Feature Consistency via wavelet decomposition. These components collectively enable robust spatiotemporal feature alignment in challenging underwater environments.

---

**Algorithm 1** Diffusion-based Feature Correction (DFC)

---

**Require:** Input features $\mathbf{X} \in \mathbb{R}^{B \times L \times C}$, diffusion step $t$ (default: $T = 700$)
**Ensure:** Corrected features $\widetilde{\mathbf{X}}$
1: $\mathbf{H} \leftarrow \text{LayerNorm}\left(\mathbf{W}_{p2} \cdot \text{GELU}\left(\mathbf{W}_{p1} \cdot \mathbf{X}\right)\right)$
2: $\mathbf{c} \leftarrow \mathbf{W}_{c2} \cdot \text{GELU}\left(\mathbf{W}_{c1} \cdot \left(\frac{1}{L} \sum_{i=1}^{L} \mathbf{H}_i\right)\right)$
3: $\mathbf{t}_{\text{emb}} \leftarrow \text{LayerNorm}\left(\left[\sin\left(t \cdot e^{-k}\right), \cos\left(t \cdot e^{-k}\right)\right]_{k=0}^{D/2-1}\right)$
4: $\eta \leftarrow \cos\left(\frac{\pi}{2} \cdot \frac{t+s}{T+s}\right)$
5: $\mathbf{H}_{\text{corr}} \leftarrow f_\theta(\mathbf{H}, \mathbf{t}_{\text{emb}}, \mathbf{c})$
6: $\mathbf{H}'_{\text{corr}} \leftarrow \mathbf{H}_{\text{corr}} \cdot (1 + \eta) + \lambda \cdot \epsilon$ where $\epsilon \sim \mathcal{N}(0, \mathbf{I})$
7: $\mathbf{g} \leftarrow \sigma\left(\mathbf{W}_g \cdot [\mathbf{H} \oplus \mathbf{H}'_{\text{corr}}]\right)$
8: $\widetilde{\mathbf{X}} \leftarrow \mathbf{g} \odot \mathbf{H}'_{\text{corr}} + (1 - \mathbf{g}) \odot \mathbf{H}$

---

**Diffusion-based Feature Correction Algorithm** Algorithm 1 implements the diffusion-based feature correction process to address underwater degradation, as detailed in Section 3.2. The process begins by transforming input features $\mathbf{X}$ through a multi-layer perceptron (MLP) with layer normalization, producing enriched representations $\mathbf{H}$ via $\mathbf{H} = \text{LayerNorm}\left(\mathbf{W}_{p2} \cdot \text{GELU}\left(\mathbf{W}_{p1} \cdot \mathbf{X}\right)\right)$, where $\mathbf{W}_{p1} \in \mathbb{R}^{C \times 2C}$ and $\mathbf{W}_{p2} \in \mathbb{R}^{2C \times C}$ are learnable weights. This step enhances feature expressiveness and stability for subsequent processing.

A condition vector $\mathbf{c}$ is then generated to encapsulate global contextual information by averaging $\mathbf{H}$ over the sequence dimension and applying another MLP: $\mathbf{c} = \mathbf{W}_{c2} \cdot \text{GELU}\left(\mathbf{W}_{c1} \cdot \left(\frac{1}{L} \sum_{i=1}^{L} \mathbf{H}_i\right)\right)$, with $\mathbf{W}_{c1} \in \mathbb{R}^{C \times F}$ and $\mathbf{W}_{c2} \in \mathbb{R}^{F \times F}$. This vector guides the diffusion process by providing a consistent reference of the sequence's overall style, crucial for maintaining temporal coherence against color distortion.

For the diffusion process, a time step embedding $\mathbf{t}_{\text{emb}}$ is computed using sinusoidal encoding and layer normalization to represent temporal information continuously. Simultaneously, a noise factor $\eta$ is derived from a cosine schedule: $\eta = \cos\left(\frac{\pi}{2} \cdot \frac{t+s}{T+s}\right)$, where $T = 700$ and $s$ is a small constant. The core correction is performed by a U-Net architecture $f_\theta$ that takes $\mathbf{H}$, $\mathbf{t}_{\text{emb}}$, and $\mathbf{c}$ as inputs, outputting corrected features $\mathbf{H}_{\text{corr}}$. This approximates the reverse diffusion step, theoretically described by $\mathbf{H}_{t-1} = \frac{1}{\sqrt{\alpha_t}}\left(\mathbf{H}_t - \frac{1-\alpha_t}{\sqrt{1-\gamma_t}} f_\theta(\mathbf{c}, \mathbf{H}_t, t)\right) + \lambda_t \beta_t \epsilon_t$.

To enhance robustness, noise is added to $\mathbf{H}_{\text{corr}}$: $\mathbf{H}'_{\text{corr}} = \mathbf{H}_{\text{corr}} \cdot (1 + \eta) + \lambda \cdot \epsilon$, where $\epsilon \sim \mathcal{N}(0, \mathbf{I})$ and $\lambda$ is a learnable weight. This simulates stochasticity to handle uncertainties like low visibility.

Finally, a gating mechanism fuses the original and corrected features: $\mathbf{g} = \sigma\left(\mathbf{W}_g \cdot [\mathbf{H} \oplus \mathbf{H}'_{\text{corr}}]\right)$ and $\widetilde{\mathbf{X}} = \mathbf{g} \odot \mathbf{H}'_{\text{corr}} + (1 - \mathbf{g}) \odot \mathbf{H}$, balancing consistency and enhancement. This approach effectively mitigates color distortion and maintains structural coherence through adaptive feature transformation.

---

**Algorithm 2** Short-Range Temporal Feature Consistency (SRTFC)

---

**Require:** Original features $\mathbf{X} \in \mathbb{R}^{B \times L \times C}$, corrected features $\widetilde{\mathbf{X}} \in \mathbb{R}^{B \times L \times C}$
**Ensure:** Style-aligned features $\mathbf{X}' \in \mathbb{R}^{B \times L \times C}$

1: $\boldsymbol{\mu} \leftarrow \frac{1}{L} \sum_{l=1}^{L} \mathbf{X}_{:,l,:}$
2: $\mathbf{s}_{\text{curr}} \leftarrow \text{ReLU}\left(\boldsymbol{\mu}\mathbf{W}_1 + \mathbf{b}_1\right) \{\mathbf{W}_1 \in \mathbb{R}^{C \times C}, \mathbf{b}_1 \in \mathbb{R}^C\}$
3: Update historical style memory $\mathbf{M} \in \mathbb{R}^{M \times C}$ with $\mathbf{s}_{\text{curr}}$ (replace oldest entry)
4: **for** each $\mathbf{s}^{(k)} \in \mathbf{M}$ **do**
5: $\quad c_k \leftarrow \frac{\mathbf{s}_{\text{curr}} \cdot \mathbf{s}^{(k)}}{\|\mathbf{s}_{\text{curr}}\| \|\mathbf{s}^{(k)}\|}$
6: **end for**
7: Select top-$K$ indices with highest $c_k$
8: $\omega_k \leftarrow \frac{\exp(c_k)}{\sum_{j=1}^{K} c_j)}$ for $k = 1$ to $K$
9: $\mathbf{s}_{\text{fused}} \leftarrow \sum_{k=1}^{K} \omega_k \mathbf{s}^{(k)}$
10: Apply instance normalization to $\widetilde{\mathbf{X}}$:
11: $\quad$ For each $b$ and $c$: $\mu_{b,c} \leftarrow \frac{1}{L} \sum_{l=1}^{L} \widetilde{X}_{b,l,c}, \sigma_{b,c}^2 \leftarrow \frac{1}{L} \sum_{l=1}^{L} (\widetilde{X}_{b,l,c} - \mu_{b,c})^2$
12: $\quad \hat{X}_{\text{norm},b,l,c} \leftarrow \frac{\widetilde{X}_{b,l,c} - \mu_{b,c}}{\sqrt{\sigma_{b,c}^2 + \epsilon}} \cdot \gamma_c + \beta_c \{\gamma_c, \beta_c \text{ are learnable parameters}\}$
13: $\mathbf{X}_{\text{transferred}} \leftarrow \text{LinearInterpolation}(\hat{\mathbf{X}}_{\text{norm}}, \mathbf{s}_{\text{fused}})$ {Details omitted as per method section}
14: $\mathbf{X}_{\text{smooth}} \leftarrow \text{Conv1D}(\mathbf{X}_{\text{transferred}}, \text{kernel size} = 3, \text{padding} = 1)$ {Learnable kernel $\mathbf{K}$}
15: $\mathbf{X}' \leftarrow \alpha \mathbf{X}_{\text{transferred}} + (1 - \alpha)\mathbf{X}_{\text{smooth}} \{\alpha = 0.7\}$

---

**Short-Range Temporal Feature Consistency Algorithm** Algorithm 2 implements the short-range temporal feature consistency mechanism, as described in Section 3.3. The process starts by extracting the current style vector $\mathbf{s}_{\text{curr}}$ from the original features $\mathbf{X}$: $\boldsymbol{\mu} = \frac{1}{L} \sum_{l=1}^{L} \mathbf{X}_{:,l,:}$ and $\mathbf{s}_{\text{curr}} = \text{ReLU}\left(\boldsymbol{\mu}\mathbf{W}_1 + \mathbf{b}_1\right)$, where $\mathbf{W}_1 \in \mathbb{R}^{C \times C}$ and $\mathbf{b}_1 \in \mathbb{R}^C$. This vector captures the dominant stylistic characteristics of the current input.

A dynamic memory matrix $\mathbf{M} \in \mathbb{R}^{M \times C}$ stores historical style vectors and is updated by replacing the oldest entry with $\mathbf{s}_{\text{curr}}$, ensuring relevance. To fuse historical styles, the cosine similarity between $\mathbf{s}_{\text{curr}}$ and each $\mathbf{s}^{(k)}$ in $\mathbf{M}$ is computed: $\cos(\mathbf{s}_{\text{curr}}, \mathbf{s}^{(k)}) = \frac{\mathbf{s}_{\text{curr}} \cdot \mathbf{s}^{(k)}}{\|\mathbf{s}_{\text{curr}}\| \|\mathbf{s}^{(k)}\|}$. The top-$K$ most similar styles are selected, and their weights are obtained via softmax: $\omega_k = \frac{\exp(\cos(\mathbf{s}_{\text{curr}}, \mathbf{s}^{(k)}))}{\sum_{j=1}^{K} \exp(\cos(\mathbf{s}_{\text{curr}}, \mathbf{s}^{(j)}))}$, yielding a fused style vector $\mathbf{s}_{\text{fused}} = \sum_{k=1}^{K} \omega_k \mathbf{s}^{(k)}$.

Style adaptive normalization is then applied to the diffusion-corrected features $\widetilde{\mathbf{X}}$. Instance normalization is performed per channel: for each batch $b$ and channel $c$, $\mu_{b,c} = \frac{1}{L} \sum_{l=1}^{L} \widetilde{X}_{b,l,c}$ and $\sigma_{b,c}^2 = \frac{1}{L} \sum_{l=1}^{L} (\widetilde{X}_{b,l,c} - \mu_{b,c})^2$, resulting in $\hat{X}_{\text{norm},b,l,c} = \frac{\widetilde{X}_{b,l,c} - \mu_{b,c}}{\sqrt{\sigma_{b,c}^2 + \epsilon}} \cdot \gamma_c + \beta_c$ with learnable $\gamma_c$ and $\beta_c$. These normalized features are combined with $\mathbf{s}_{\text{fused}}$ through linear interpolation to produce style-transferred features $\mathbf{X}_{\text{transferred}}$.

Temporal smoothing enhances coherence across sequences using a 1D convolution with a learnable kernel $\mathbf{K} \in \mathbb{R}^3$ and padding of 1: $X_{\text{smooth},b,l,c} = \sum_{d=-1}^{1} K_{d+1} \cdot X_{\text{transferred},b,l+d,c}$. The output features are formed by a weighted sum: $\mathbf{X}' = \alpha \mathbf{X}_{\text{transferred}} + (1 - \alpha)\mathbf{X}_{\text{smooth}}$ with $\alpha = 0.7$, balancing feature integrity and temporal smoothness. This ensures style consistency and coherence in underwater tracking.

**Long-Range Temporal Feature Consistency Algorithm** Algorithm 3 implements the long-range temporal feature consistency mechanism based on ODTrack, as detailed in Section 3.4. The input consists of the previous token sequence $\mathbf{T}_{t-1}$ and current token sequence $\mathbf{T}_t$, both padded to a

---

**Algorithm 3** Long-Range Temporal Feature Consistency (LRTFC)

---

**Require:** Previous token sequence $\mathbf{T}_{t-1} \in \mathbb{R}^{B \times N \times C}$, current token sequence $\mathbf{T}_t \in \mathbb{R}^{B \times N \times C}$

**Ensure:** Propagated token sequence $\mathbf{T}_{\text{out}} \in \mathbb{R}^{B \times N \times C}$

1: Pad $\mathbf{T}_{t-1}$ and $\mathbf{T}_t$ to minimum length $N_{\min} = 4$ if $N < 4$ using zero-padding
2: Reshape $\mathbf{T}_{t-1}$ to $\mathbf{T}_{t-1}^{\text{2D}} \in \mathbb{R}^{B \times C \times H \times W}$ where $H \times W = N$
3: Apply 2D discrete wavelet transform (Haar) to obtain $\mathbf{T}_{LL}, \mathbf{T}_{LH}, \mathbf{T}_{HL}, \mathbf{T}_{HH} \in \mathbb{R}^{B \times C \times H/2 \times W/2}$
4: Reshape components back to token format $\mathbb{R}^{B \times N' \times C}$ and truncate to length $N$
5: Concatenate components: $\mathbf{X} \leftarrow [\mathbf{T}_{LL}, \mathbf{T}_{LH}, \mathbf{T}_{HL}, \mathbf{T}_{HH}, \mathbf{T}_t] \in \mathbb{R}^{B \times N \times 5C}$
6: Compute fusion weights: $\mathbf{G} \leftarrow \sigma\left(\mathbf{W}_g \mathbf{X} + \mathbf{b}_g\right)$ where $\mathbf{W}_g \in \mathbb{R}^{5C \times 2C}$, $\mathbf{b}_g \in \mathbb{R}^{2C}$
7: Split $\mathbf{G}$ into $\mathbf{G}_1 \in \mathbb{R}^{B \times N \times C}$ and $\mathbf{G}_2 \in \mathbb{R}^{B \times N \times C}$
8: Sum high-frequency components: $\mathbf{T}_{\text{HF}} \leftarrow \mathbf{T}_{LH} + \mathbf{T}_{HL} + \mathbf{T}_{HH}$
9: Modulate high-frequency: $\mathbf{T}_{\text{high}} \leftarrow \mathbf{T}_{\text{HF}} \odot \mathbf{G}_1$
10: Combine low-frequency and current token: $\mathbf{T}_{\text{new\_ll}} \leftarrow \mathbf{T}_{LL} \odot \mathbf{G}_2 + \mathbf{T}_t \odot (\mathbf{1} - \mathbf{G}_2)$
11: Concatenate results: $\mathbf{T}_{\text{fused}} \leftarrow [\mathbf{T}_{\text{new\_ll}}, \mathbf{T}_{\text{high}}] \in \mathbb{R}^{B \times N \times 2C}$
12: Project to original dimension: $\mathbf{T}_{\text{out}} \leftarrow \mathbf{W}_p \mathbf{T}_{\text{fused}} + \mathbf{b}_p$ where $\mathbf{W}_p \in \mathbb{R}^{2C \times C}$, $\mathbf{b}_p \in \mathbb{R}^C$

---

minimum length of 4 tokens if necessary using zero-padding to ensure compatibility with wavelet transformation.

The previous token sequence $\mathbf{T}_{t-1}$ is reshaped into a 2D format $\mathbf{T}_{t-1}^{\text{2D}} \in \mathbb{R}^{B \times C \times H \times W}$ where $H \times W = N$. A 2D discrete wavelet transform (DWT) with Haar wavelet decomposes it into low-frequency component $\mathbf{T}_{LL} \in \mathbb{R}^{B \times C \times H/2 \times W/2}$ and high-frequency components $\mathbf{T}_{LH}, \mathbf{T}_{HL}, \mathbf{T}_{HH} \in \mathbb{R}^{B \times C \times H/2 \times W/2}$. These components are reshaped back to token format and truncated to the original length $N$.

The components are concatenated with the current token $\mathbf{T}_t$ along the feature dimension: $\mathbf{X} = [\mathbf{T}_{LL}, \mathbf{T}_{LH}, \mathbf{T}_{HL}, \mathbf{T}_{HH}, \mathbf{T}_t] \in \mathbb{R}^{B \times N \times 5C}$. Fusion weights are generated via a linear transformation and sigmoid activation: $\mathbf{G} = \sigma\left(\mathbf{W}_g \mathbf{X} + \mathbf{b}_g\right)$ with $\mathbf{W}_g \in \mathbb{R}^{5C \times 2C}$ and $\mathbf{b}_g \in \mathbb{R}^{2C}$. The weights are split into $\mathbf{G}_1$ and $\mathbf{G}_2$.

The high-frequency components are summed: $\mathbf{T}_{\text{HF}} = \mathbf{T}_{LH} + \mathbf{T}_{HL} + \mathbf{T}_{HH}$, and modulated by $\mathbf{G}_1$ through element-wise multiplication: $\mathbf{T}_{\text{high}} = \mathbf{T}_{\text{HF}} \odot \mathbf{G}_1$. The low-frequency component and current token are combined using $\mathbf{G}_2$: $\mathbf{T}_{\text{new\_ll}} = \mathbf{T}_{LL} \odot \mathbf{G}_2 + \mathbf{T}_t \odot (\mathbf{1} - \mathbf{G}_2)$. The results are concatenated: $\mathbf{T}_{\text{fused}} = [\mathbf{T}_{\text{new\_ll}}, \mathbf{T}_{\text{high}}] \in \mathbb{R}^{B \times N \times 2C}$, and projected back to the original dimension via a linear layer: $\mathbf{T}_{\text{out}} = \mathbf{W}_p \mathbf{T}_{\text{fused}} + \mathbf{b}_p$ with $\mathbf{W}_p \in \mathbb{R}^{2C \times C}$ and $\mathbf{b}_p \in \mathbb{R}^C$. This ensures temporal coherence and enhances discrimination against distractors in underwater environments.

# D ADDITIONAL EXPERIMENTS

Table 6: Comparison of different feature correction methods on VMAT.

| Method | AUC | P | Params | FPS |
|---|---|---|---|---|
| *Diffusion* | 61.25 | 78.07 | 98.70M | 32.35 |
| *GAN* | 56.04 | 70.98 | 111.13M | 23.62 |
| *Transformer* | 57.41 | 72.07 | 100.52M | 28.23 |

Table 7: Compare the historical style area size of SRTFC on VMAT.

| Size | AUC | P |
|---|---|---|
| 4 | 60.99 | 77.70 |
| 5 | 61.25 | 78.07 |
| 6 | 60.20 | 76.26 |

**Different feature correction methods** Table 6 demonstrate the superior performance of our proposed diffusion-based feature correction method on the VMAT dataset. With an AUC of 61.25% and precision of 78.07%, our approach outperforms both GAN-based (56.04% AUC, 70.98% precision) and Transformer-based (57.41% AUC, 72.07% precision) alternatives. This performance advantage is attributed to the progressive denoising mechanism of the diffusion process, which effectively handles feature degradation in underwater environments through multi-step iterative refinement. The method also maintains practical efficiency with 98.70M parameters and a processing speed of 32.35

FPS, striking an optimal balance between model complexity and computational requirements. The conditional guidance mechanism ensures temporal coherence while preventing fragmentation issues common in frame-independent processing, making it particularly suitable for challenging underwater tracking scenarios where feature consistency is crucial for robust performance.

**Historical memory size** The ablation results in Table 7 demonstrate that a historical style area size of 5 achieves optimal performance with 61.25% AUC and 78.07% precision on VMAT, outperforming both smaller (60.99% AUC, 77.7% precision) and larger (60.20% AUC, 76.26% precision) configurations. This optimal size effectively balances temporal information utilization with noise suppression through the dynamic style memory mechanism that selects the most relevant historical styles based on cosine similarity, validating the design rationale of the Short-Range Temporal Feature Consistency (SRTFC) for underwater object tracking.

## E    ADDITIONAL VISUALIZATIONS

Comprehensive visual analyses validate UTFC-DiffTracker's performance. Attention maps demonstrate feature refinement across three modules. Qualitative comparisons show robustness against four underwater challenges. Detailed visualizations illustrate key scenarios including color distortion, low visibility, similar distractors, and full occlusion. These analyses collectively confirm the framework's spatiotemporal alignment capabilities.

**Qualitative Comparison.** Figure 6 demonstrates UTFC-DiffTracker's robustness across four underwater challenges on WebUOT-1M. The integrated framework maintains precise tracking where competitors fail: sustaining through frame 390 in color distortion while others fail by frame 384; tracking until frame 1727 in low visibility while competitors lose before; maintaining accuracy through frame 204 against distractors while others fail by frame 109; and recovering rapidly post-occlusion while competitors show degradation. These comparisons validate the framework's correction mechanisms.

**Attention Visualization.** Figure 7 demonstrates attention refinement across three modules and Figure 8 presents the visualization of the tracking results among the three modules. The Diffusion-based Feature Correction (DFC) reduces attention dispersion under color distortion. Without DFC (column 1), attention scatters across non-target regions. With DFC (column 2), attention converges on target boundaries. This shows effective feature correction through adaptive gating. The Short-Range Temporal Feature Consistency (SRTFC) resolves temporal fragmentation. Without SRTFC (column 3), attention weights vary between frames 128-129. With SRTFC (column 4), attention maintains coherent focus. This achieves temporal consistency through style retention. The Long-Range Temporal Feature Consistency (LRTFC) enhances distractor discrimination. Without LRTFC (column 5), attention leaks to background regions. With LRTFC (column 6), attention suppresses distractors. Wavelet decomposition preserves structural components. These visualizations validate spatiotemporal alignment for reliable underwater tracking.

Detailed visualizations for each challenge are provided in Figures 9-12. Color distortion challenges show severe blending in greenish water (Figure 9a) and low visibility in bluish-green water (Figure 9b). Low visibility scenarios include dark environments (Figure 10a) and turbid water with small targets (Figure 10b). Similar distractor cases feature dense fish schools (Figure 11a) and uniform dark fish (Figure 11b). Full occlusion demonstrates sudden blocking (Figure 12a) and gradual covering (Figure 12b). UTFC-DiffTracker consistently outperforms competitors across all scenarios.

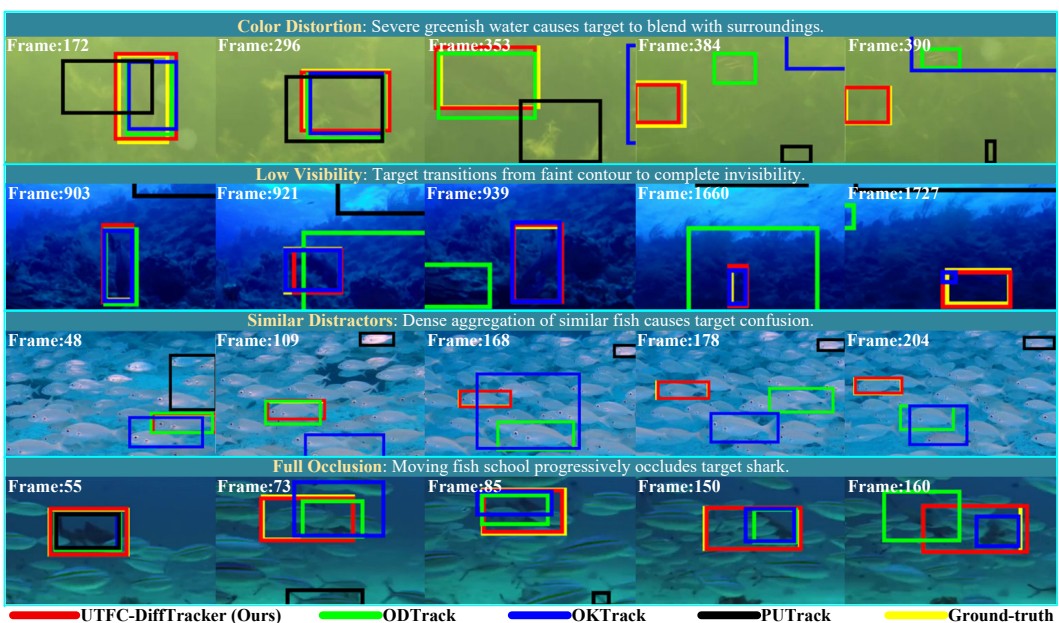

Figure 6: Comprehensive qualitative results on WebUOT-1M showing robustness to four underwater challenges.

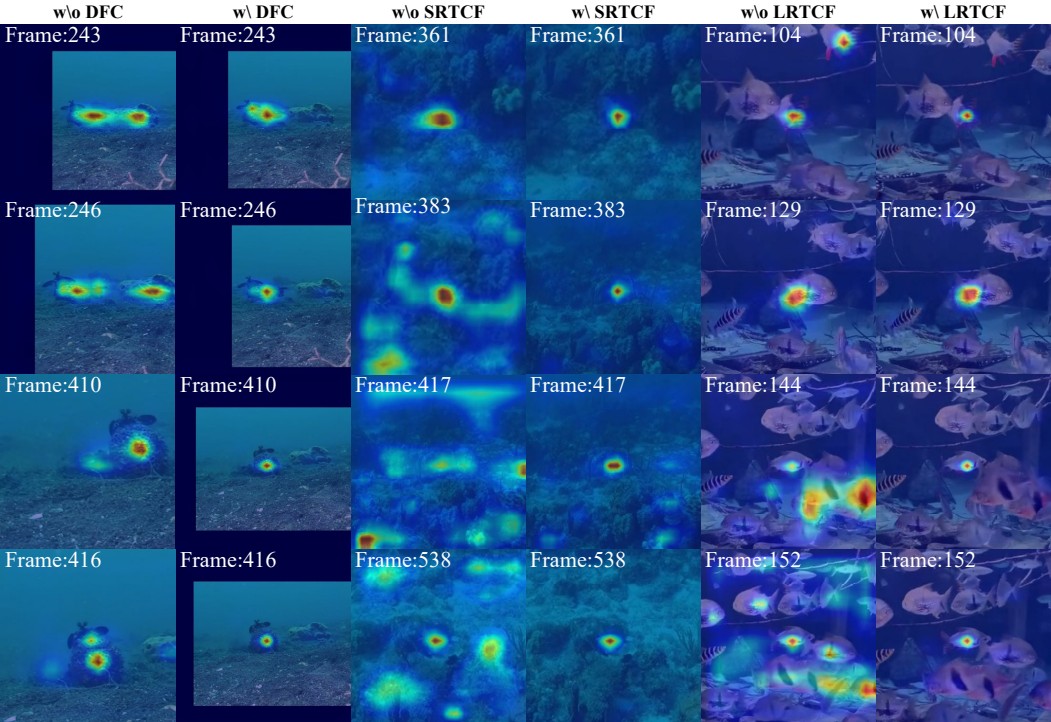

Figure 7: Attention maps for DFC, STRCF, and LRTFC modules. Rows show trackers without (w/o) and with (w/) each module.

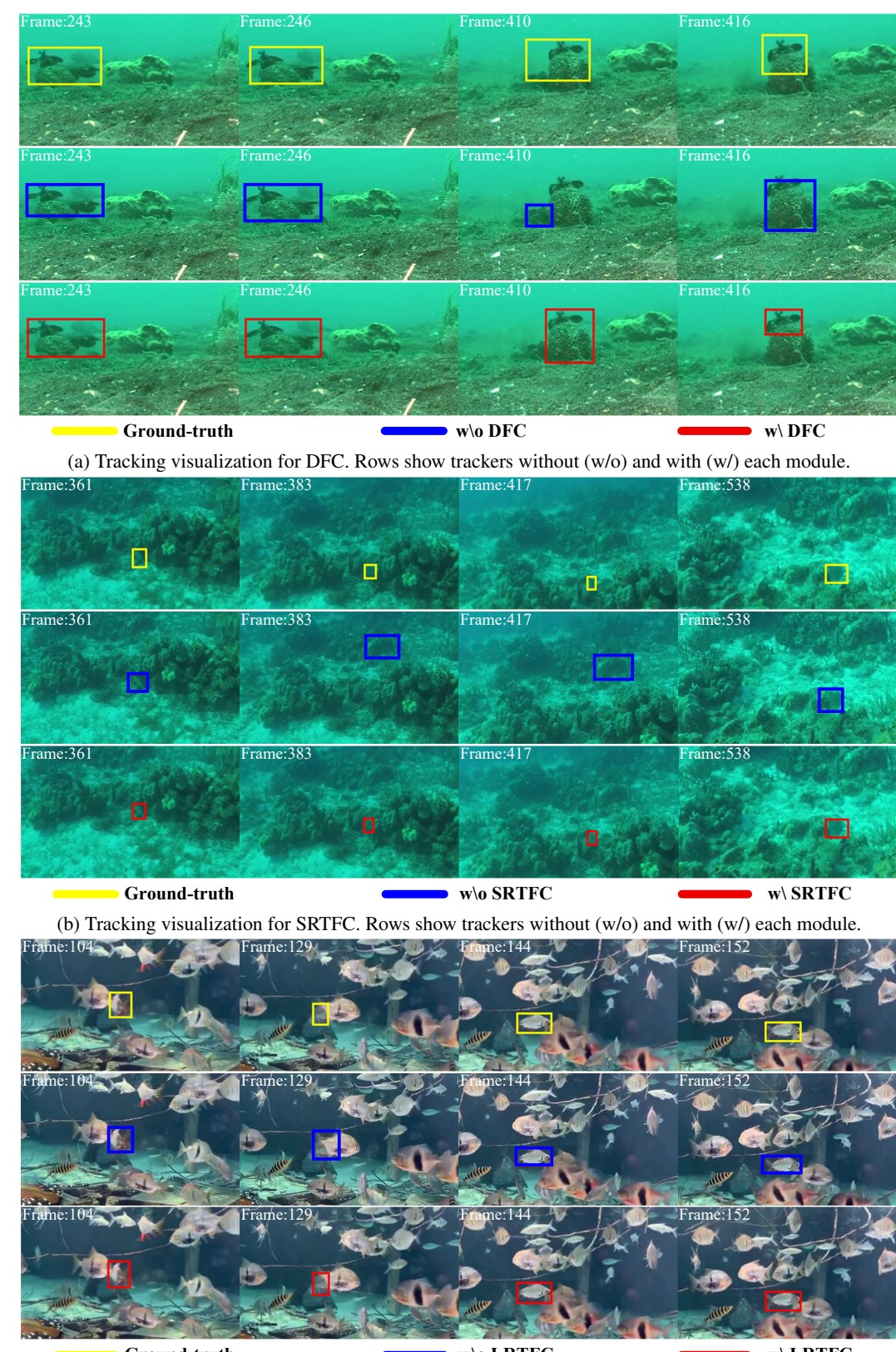

(a) Tracking visualization for DFC. Rows show trackers without (w/o) and with (w/) each module.

(b) Tracking visualization for SRTFC. Rows show trackers without (w/o) and with (w/) each module.

(c) Tracking visualization for LRTFC. Rows show trackers without (w/o) and with (w/) each module.

Figure 8: Visualization of the tracking results among the three modules.

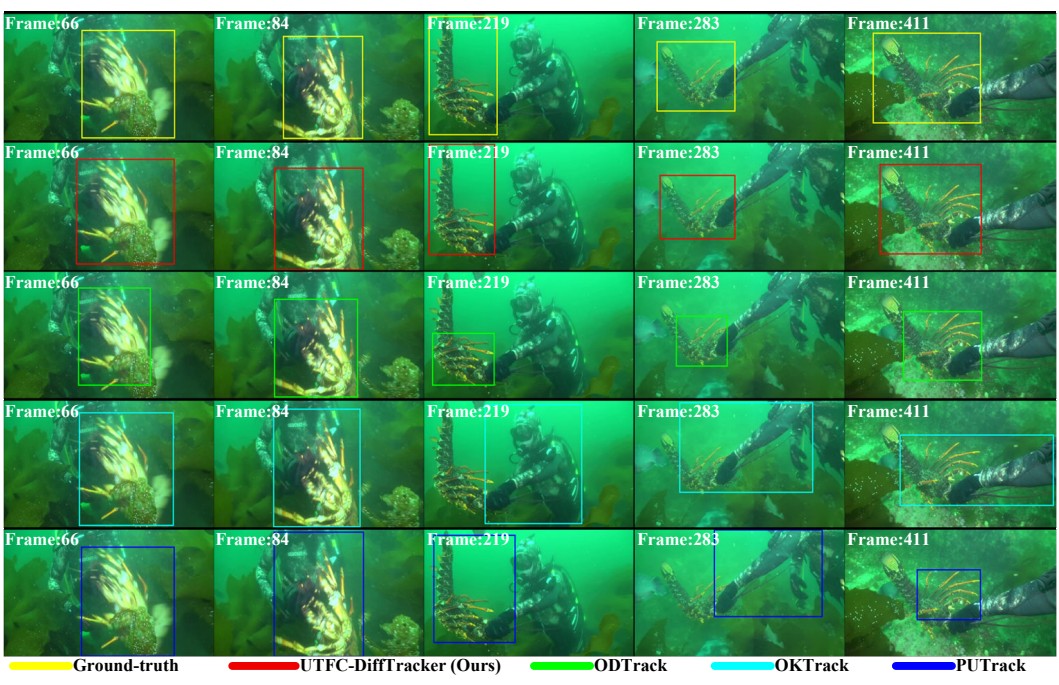

(a) Diver capturing lobster in greenish water: severe color distortion blends target with surroundings.

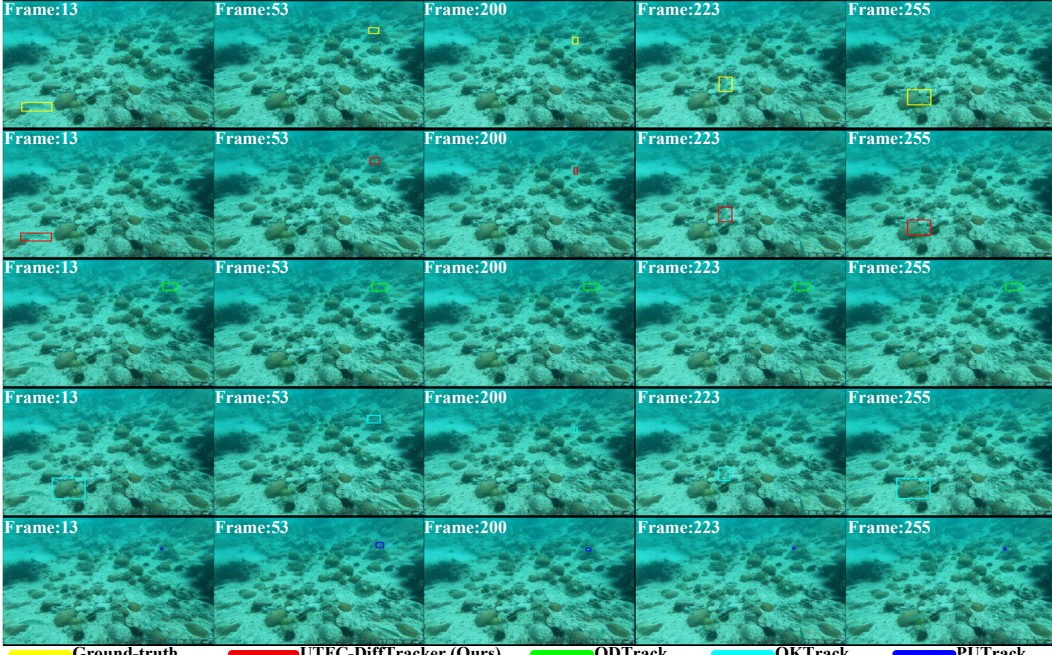

(b) Fish movement in bluish-green water: color distortion causes low visibility and blending.

Figure 9: Color distortion challenges on WebUOT-1M benchmark.

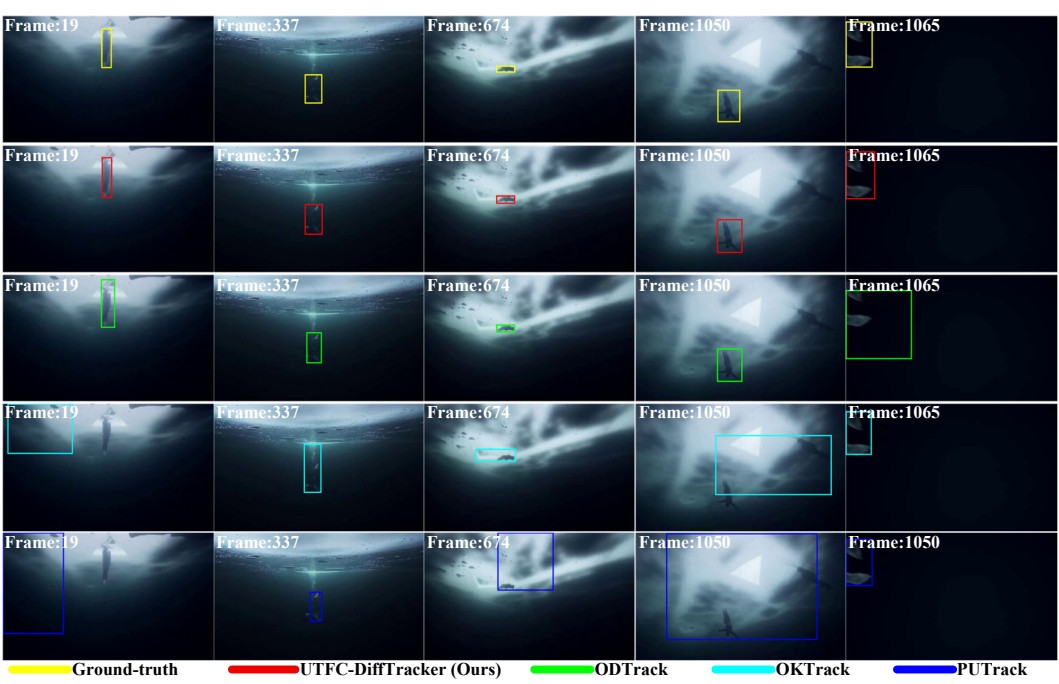

(a) Diver in low-light environment: extremely poor visibility challenges tracking.

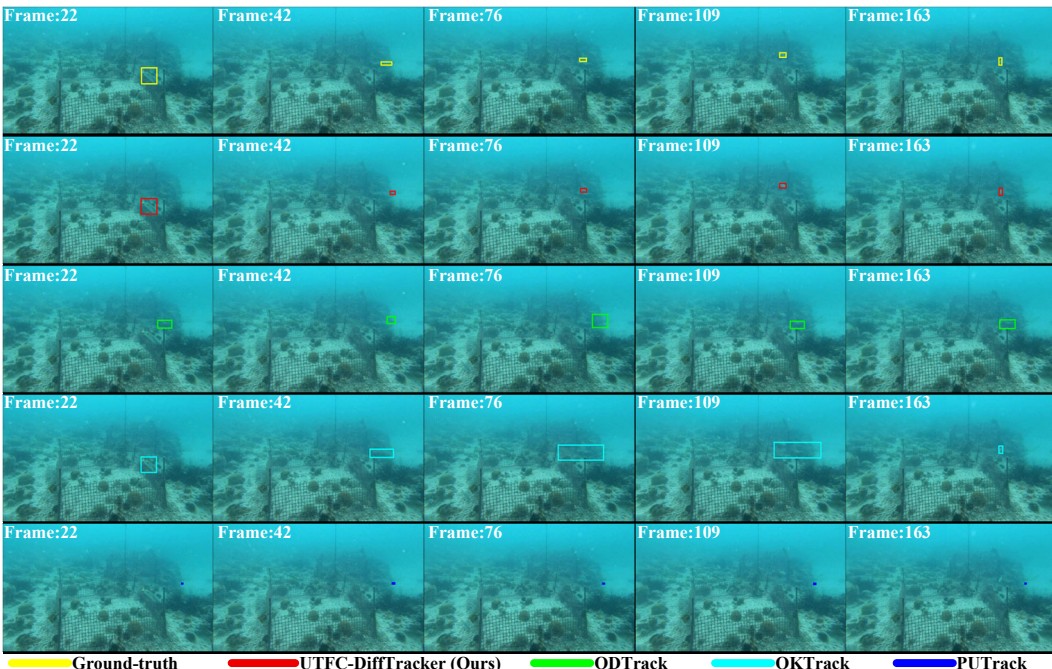

(b) Tiny fish in turbid water: low visibility and small size hinder detection.

Figure 10: Low visibility challenges on WebUOT-1M benchmark.

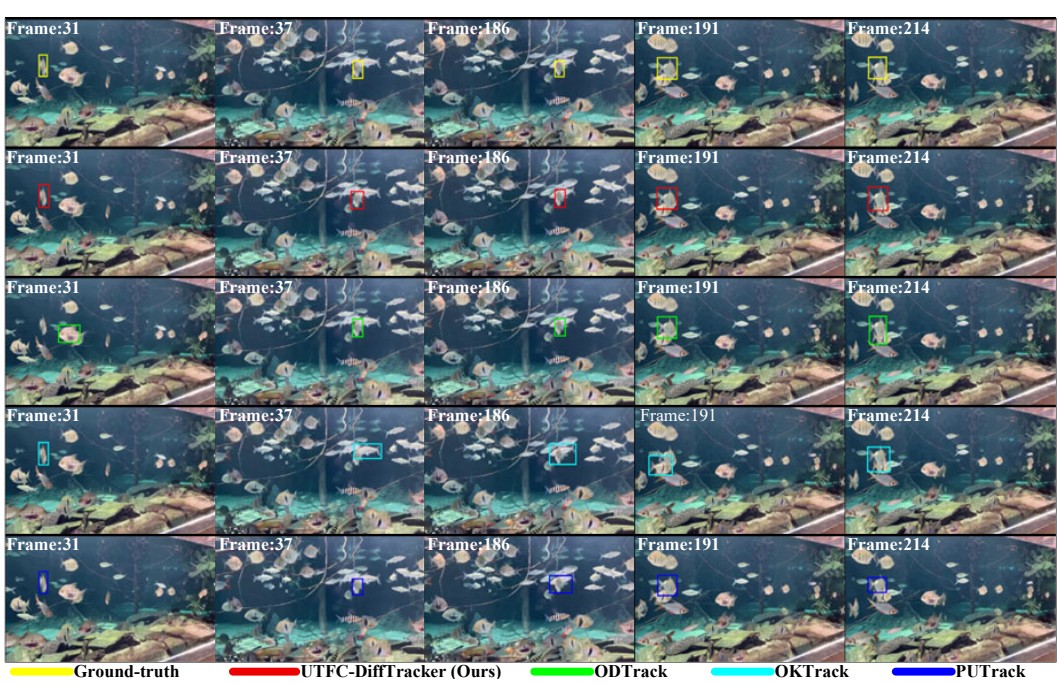

(a) Fish tracking in dense school: similar distractors cause severe interference.

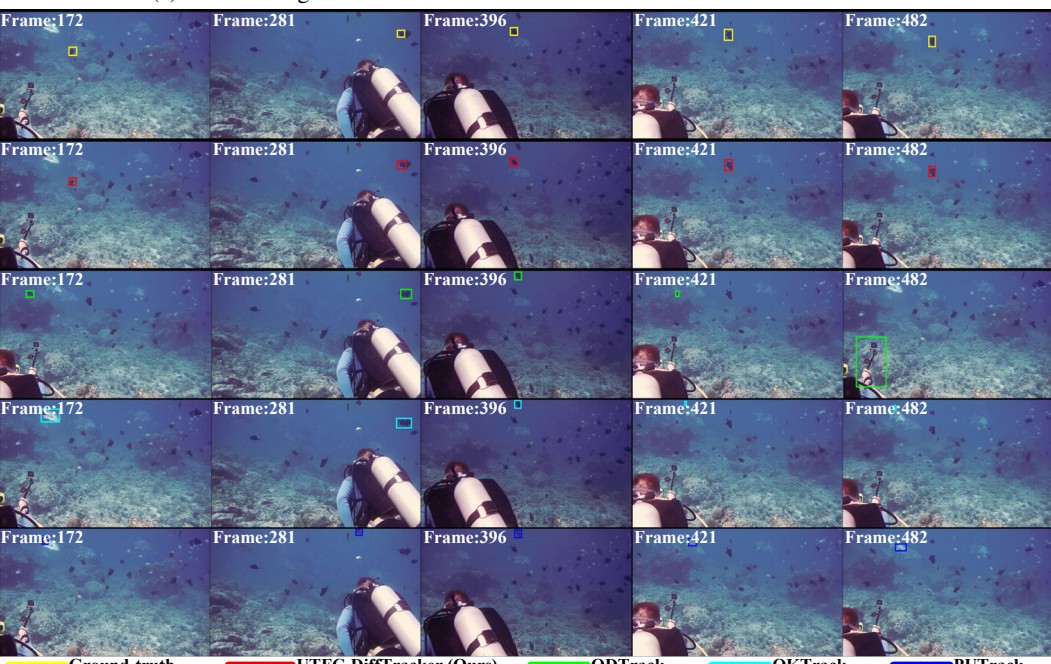

(b) Dark fish in uniform school: identical appearance induces tracking drift.

Figure 11: Similar distractor challenges on WebUOT-1M benchmark.

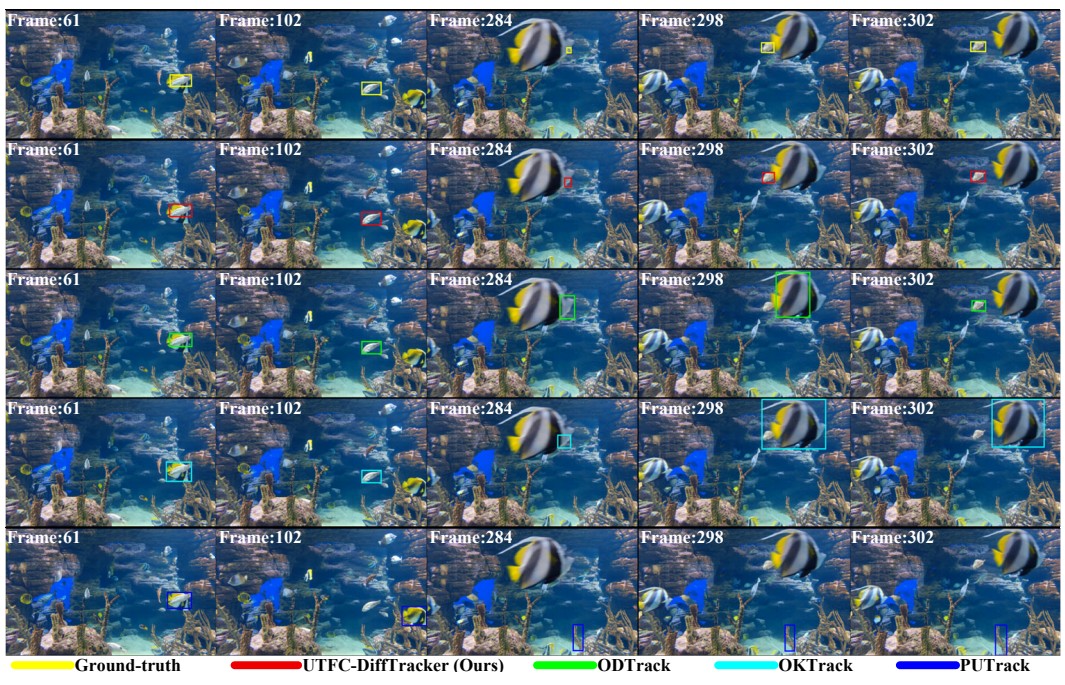

(a) Sudden fish occlusion: passing fish completely blocks target view.

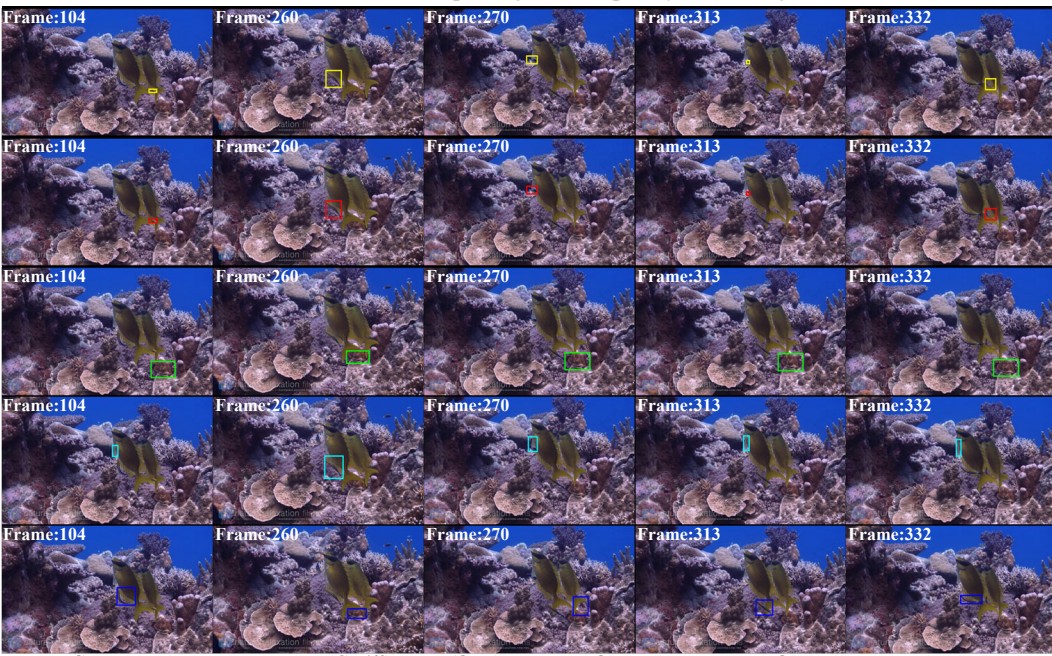

(b) Gradual occlusion by similar fish: target becomes fully blocked.

Figure 12: Full occlusion challenges on WebUOT-1M benchmark.

