# OpenReview forum: "UTFC-DiffTracker: Short- and Long-Range Temporal Feature Consistency Diffusion for Underwater Object Tracking"
_ICLR.cc/2026/Conference — ICLR 2026 Conference Withdrawn Submission_

### Official Review · Reviewer_Wdoh · 2025-10-30

**Soundness:** 3
**Presentation:** 2
**Contribution:** 3
**Rating:** 4
**Confidence:** 3

**Summary:**

The paper proposes a new tracking framework designed to address the unique challenges of underwater object tracking, such as color distortion, low visibility, similar distractors, and occlusion.

**Strengths:**

1.The experiments are thorough, and the performance improvements are significant.

2.The proposed Short-Range Temporal Feature Consistency module and Long-Range Temporal Feature Consistency module can, to some extent, address the unique challenges of underwater object tracking.

**Weaknesses:**

1.Video-level trackers suffer from feature degradation problems, but the authors propose that their feature-level tracker can effectively address these issues. Why is that?

2.The proposed method can improve tracking performance in underwater tasks to some extent, but its theoretical innovation remains insufficient. It appears more like a combination of Model A and Model B.

3.In Figure 2, what do “Style Fusion,” “Smooth,” and “Fusion” mean, and how are they designed internally? The overall design of Figure 2 doesn’t look very academic, which makes it difficult for me to assess the innovativeness of the paper.

4.The authors only present successful cases in their visualizations. I would like to know whether there are any failure cases, and under what circumstances such results might occur.

**Questions:**

1.In Figure 1, it is not clearly indicated which specific methods produced the results shown in parts (a) and (b).

2.Regarding the issue of fairness, did the authors use the same training dataset and train all comparison methods for 300 epochs? Alternatively, did the authors provide an explanation for this?

3.For the second question, if the authors provide a reasonable explanation, I will raise my score.

---

### Official Review · Reviewer_VbiF · 2025-10-31

**Soundness:** 2
**Presentation:** 1
**Contribution:** 2
**Rating:** 2
**Confidence:** 5

**Summary:**

This paper proposes a tracking algorithm called UTFC-DiffTracker for underwater target tracking. The authors summarize the contributions of their work into two main modules: Short-Range Temporal Feature Consistency (SRTFC) and Long-Range Temporal Feature Consistency (LRTFC). SRTFC is based on a diffusion model to obtain additional features for feature fusion, but it lacks effective supervision signals. The temporal modeling mechanism of LRTFC is derived from ODTrack, with the introduction of a 2D discrete wavelet transform (DWT) signal on this basis.

**Strengths:**

1.The experiments are solid and comprehensive. The authors conducted evaluations on four existing public underwater tracking datasets (WebUOT-1M, UW-COT220, VMAT, and UTB180), which demonstrate the effectiveness of the proposed method.

2.The idea of short and long-range temporal feature modeling offers certain inspiration for underwater target tracking.

3.The authors clearly provide implementation details and present comprehensive ablation experiments.

**Weaknesses:**

The method proposed in this paper demonstrates some innovation, but the presentation is unsatisfactory. For example, the motivation, as well as the visualizations and textual descriptions of the approach, are unclear, insufficient, and even misleading.

1.The motivation needs to be further strengthened and made more explicit. Since the method in this paper is based on the video-based approach ODTrack, it is necessary to further clarify the limitations of existing video-based methods and explain why the proposed method can address these issues.

2.The Section 2 "Related Work" is overly simplistic and lacks systematic organization. The authors are advised to refer to existing survey literature for restructuring, in order to demonstrate the developmental landscape of previous methods and highlight the technical contributions of the proposed approach.

3.The Short-Range Temporal Feature Consistency module comprises two parts: Diffusion-based Feature Correction (DCF) and Short-Range Temporal Feature Consistency (SRTFC). However, the related textual descriptions in Section 3 Method and Figure 2 lack connection and are confusing. More seriously, a large number of details are not shown in Figure 2 (a)-(d).

4.Some statements lack experimental verification. For example, "Diffusion-based Feature Correction (DFC) adaptively transforms features to mitigate color distortion and low visibility."

5.The authors did not compare with recent diffusion-based and mamba-based tracking algorithms.

6.Furthermore, there is no detailed description of the diffusion module. It is unclear whether it uses the original stable diffusion or DiffusionDet.

**Questions:**

1.The authors are advised to further refine the motivation behind the proposed method.

2.It is recommended to rewrite Sections 2 and 3 to make the article more readable, with clearer and more accurate accompanying figures.

3.An open question: the ODTrack-based temporal token learning method is not optimal, and there are many alternative improvement schemes—why did the authors still choose it as the baseline? To my knowledge, ODTrack's training speed is very slow. It is advisable to explore more effective temporal modeling methods.

---

### Official Review · Reviewer_L59u · 2025-11-01

**Soundness:** 3
**Presentation:** 2
**Contribution:** 2
**Rating:** 4
**Confidence:** 3

**Summary:**

This paper proposes UTFC-DiffTracker, a novel framework for underwater object tracking (UOT) that aims to address challenges like color distortion, low visibility, and occlusions. The core innovation lies in combining two modules: a Short-Range Temporal Feature Consistency (SRTFC) module that uses a diffusion model and dynamic style memory to correct features and maintain frame-to-frame coherence, and a Long-Range Temporal Feature Consistency (LRTFC) module that employs wavelet decomposition on historical tokens to separate stable structures from transient details, improving robustness to distractors and occlusions. The authors report state-of-the-art results on four UOT benchmarks, with significant improvements over existing methods.

**Strengths:**

1. The proposed integration of a diffusion model for feature correction and a wavelet-based decomposition for long-term temporal modeling is creative and has not been extensively explored in the tracking literature. The dynamic style memory for short-range consistency is also a noteworthy idea.

2. The paper provides a thorough quantitative evaluation on multiple established UOT benchmarks, demonstrating a significant performance lead over a wide range of competitors. The inclusion of detailed ablation studies and component analyses (e.g., diffusion steps, memory update strategies) is a strength.

**Weaknesses:**

The paper operates more as an engineering demonstration than a scientific contribution. It fails to provide a deep, principled justification for its core design choices:

1. Diffusion Model Rationale: The use of a computationally heavy diffusion process for feature correction is not sufficiently motivated. The authors do not explain why a diffusion model is uniquely suited for this task compared to a simpler denoising autoencoder or a flow-based model. The claim that underwater degradation manifests as "predictable distribution shifts" is unsubstantiated, and the complex, iterative denoising process seems disproportionate to the problem.

2. Wavelet Decomposition Justification: The choice of the Haar wavelet is presented as an arbitrary selection that happens to work best in an ablation. There is no discussion of why wavelet analysis is a principled approach for separating "stable structures" from "transient details" in the context of tracking tokens.

**Questions:**

See weaknesses.

---

### Official Review · Reviewer_ynSc · 2025-11-01

**Soundness:** 2
**Presentation:** 3
**Contribution:** 2
**Rating:** 6
**Confidence:** 4

**Summary:**

This manuscript proposes a novel underwater target tracking method, UTFC-DiffTracker, designed to enhance tracking capability in complex underwater environments. The method integrates short-range and long-range temporal feature consistency modules and utilizes diffusion generative models to mitigate feature degradation in underwater settings while preserving temporal consistency. Through feature correction and temporal consistency, UTFC-DiffTracker achieves state-of-the-art performance across multiple underwater target tracking benchmarks.

**Strengths:**

1. UTFC-DiffTracker is a feature-aligned consistency framework that achieves spatiotemporal feature alignment and establishes new state-of-the-art results on four UOT benchmarks.
2. The paper is well-writing and organzied, with good presentation.

**Weaknesses:**

1. The manuscript proposes a cosine calculation method for noise sampling, and this hyperparameter is closely related to the step size T. Therefore, for different underwater tracking training datasets, the T value or noise value should vary significantly. How do the authors ensure the generalization of these values across different datasets.

2. To achieve LRTFC, the context relationships between video frames are densely associated using an online token propagation approach, similar to the innovation in ODTrack. It seems an incremental improvement.

3. The specific tracker limits the effects, so can it transfer to the generic object tracking task? What about the performance on LaSOT, GOT-10K and TrackingNet.

4. For the comparison in Table 1, it lacks some rencent tracking methods, such as DiffusionTrack, DeTrack, ARtrackV2, SUTrack. In addition, the ablation is incomplete, such as only SRTFC, SRTFC+LRTFC are missing.

5. As we all known, Diffusion model is time-consuming, why is there not much difference in speed before and after DFC.

6. The authors do not release the code and results. ODTrack provides a comparison of FLOPs, which is more intuitive. It is recommended to include this in the table as well.

**Questions:**

Please see weaknesses.

---

### Note · Authors · 2025-11-13

I have read and agree with the venue's withdrawal policy on behalf of myself and my co-authors.